# Plasticity of the proteasome-targeting signal Fat10 enhances substrate degradation

Hitendra Negi[1,2], Aravind Ravichandran[1,2], Pritha Dasgupta[1], Shridivya Reddy[1†], Ranabir Das[1]*

[1]National Center for Biological Sciences, Tata Institute of Fundamental Research, Bangalore, India; [2]SASTRA University, Thirumalaisamudram, Thanjavur, India

**Abstract** The proteasome controls levels of most cellular proteins, and its activity is regulated under stress, quiescence, and inflammation. However, factors determining the proteasomal degradation rate remain poorly understood. Proteasome substrates are conjugated with small proteins (tags) like ubiquitin and Fat10 to target them to the proteasome. It is unclear if the structural plasticity of proteasome-targeting tags can influence substrate degradation. Fat10 is upregulated during inflammation, and its substrates undergo rapid proteasomal degradation. We report that the degradation rate of Fat10 substrates critically depends on the structural plasticity of Fat10. While the ubiquitin tag is recycled at the proteasome, Fat10 is degraded with the substrate. Our results suggest significantly lower thermodynamic stability and faster mechanical unfolding in Fat10 compared to ubiquitin. Long-range salt bridges are absent in the Fat10 structure, creating a plastic protein with partially unstructured regions suitable for proteasome engagement. Fat10 plasticity destabilizes substrates significantly and creates partially unstructured regions in the substrate to enhance degradation. NMR-relaxation-derived order parameters and temperature dependence of chemical shifts identify the Fat10-induced partially unstructured regions in the substrate, which correlated excellently to Fat10-substrate contacts, suggesting that the tag-substrate collision destabilizes the substrate. These results highlight a strong dependence of proteasomal degradation on the structural plasticity and thermodynamic properties of the proteasome-targeting tags.

**\*For correspondence:**
rana@ncbs.res.in

**Present address:** †Institute for Stem Cell Science and Regenerative Medicine (inStem), Bangalore, India

**Competing interest:** The authors declare that no competing interests exist.

## Editor's evaluation

This manuscript probes the ways in which a protein tag might influence the structure, dynamics and stability of a covalently-attached substrate protein. These important findings are of significance to several fields, particularly in understanding how these influences control the abundance of proteins within a cell. The evidence provided is solid and the manuscript will be of interest to scientists working on protein folding and cellular degradation.

## Introduction

Proteasome substrates must be conjugated with other small proteins, known as proteasome-targeting tags, to target substrates to the proteasome (*Kwon and Ciechanover, 2017*). Ubiquitin is a proteasome-targeting tag conjugated to substrates by posttranslational modification. It interacts with the 19 S proteasome subunit, which harbors ubiquitin receptors. Consequently, 19 S aligns its base subunits, consisting of AAA +ATPases, with the 20 S Core Particle (CP) to create a channel for substrate entry. Deubiquitinase enzymes cleave the ubiquitin, while the substrate enters the ATPases to unfold and translocate to CP, where it is cleaved into short peptides by proteases (*de la Peña*

*et al., 2018*; *Wehmer et al., 2017*; *Worden et al., 2017*). Proteasomal degradation regulates most nuclear and cytosolic protein levels (*Zhao et al., 2015*). Global cellular protein degradation rate is modulated in the quiescent state, under nutrient stress, or during inflammation (*Zhao et al., 2015*; *VerPlank et al., 2019*; *Liu et al., 2021*; *Zhang et al., 2017*). However, mechanisms that regulate proteasome degradation in such conditions remain poorly understood. Partially unstructured regions at the N/C-termini or within the substrate are critical to proteasome engagement and degradation. Global disorder and topology of substrates also influence the degradation (*Correa Marrero and Barrio-Hernandez, 2021*). It is unclear if the plasticity of the proteasome-targeting tag can regulate the substrate degradation rate.

During stress or inflammation, a swift change in proteasomal degradation activity by changing the global disorder in proteins or its biased sequences may be challenging. An alternate mechanism is to upregulate a proteasome-targeting tag that rapidly degrades substrate proteins. The human leukocyte antigen-F adjacent transcript 10 (Fat10) is a proteasome-targeting tag that directly targets substrates for proteasomal degradation (*Schmidtke et al., 2014*). Fat10 expression is restricted to immune cells (*Buerger et al., 2015*), and other cell types express Fat10 upon induction by proinflammatory cytokines (*Buchsbaum et al., 2012*; *Hipp et al., 2005*; *Raasi et al., 1999*). Fat10 includes two ubiquitin-like domains attached by a flexible linker and is conjugated to substrate proteins by posttranslational modification. Fat10's surface properties and interactions are distinct from ubiquitin (*Aichem et al., 2018*). Fat10 has a lower half-life of 1 hr, while ubiquitin has a longer half-life of ~24 hr (*Haas and Bright, 1987*; *Aichem and Groettrup, 2020*). Fat10's shorter half-life prevents a prolonged inflammatory response that can potentially lead to apoptosis (*Aichem and Groettrup, 2020*). Mechanisms underlying the rapid degradation of Fat10 are poorly understood.

The substrates of Fat10 overlap significantly with ubiquitin substrates (*Aichem and Groettrup, 2020*), suggesting that Fat10 is an auxiliary proteasomal-targeting signal activated during inflammation. However, several mechanistic differences exist between the ubiquitin- and Fat10-proteasome pathways. Ubiquitin is cleaved by deubiquitinating enzymes at the proteasome before the substrate enters ATPases, but Fat10 remains uncleaved and is degraded along with the substrate (*Hipp et al., 2005*). Well-folded substrates must be globally or partially unfolded for engagement with the proteasome. Accessory unfoldases like Cdc48 assist the process by unfolding substrates (*Olszewski et al., 2019*) before they interact with the proteasome. Substrate unfolding by accessory unfoldases is necessary for the ubiquitin substrates but not for the Fat10 substrates (*Aichem et al., 2018*), suggesting a profound impact of Fat10 on the substrate's structure and energetics. However, Fat10's effect on the substrate's structure and thermodynamics is unknown.

While ubiquitin has a rigid structure, Fat10 has a ductile fold, providing an opportunity to probe if the plasticity of proteasome-targeting tags affects substrate degradation rate. In this work, we correlate the thermodynamic properties and conformational plasticity of Fat10 and ubiquitin with the proteasomal degradation rate of their substrates. Our results suggest that the Fat10 free energy barrier is substantially lower than ubiquitin. Consequently, the Fat10 unfolding kinetics are sevenfold faster than ubiquitin, and Fat10's mechanical resistance to unfolding is weaker than ubiquitin's. The absence of long-range salt bridges in Fat10 creates partially unstructured regions, leading to efficient proteasome engagement and degradation. Furthermore, Fat10's structural plasticity reduces the thermodynamic stability of substrate proteins in cellular and in vitro conditions and creates local partially unstructured regions in the substrate. The substrate reciprocally reduces Fat10 stability by thermodynamic coupling, creating more disorder in the conjugate and enhancing degradation. NMR experiments measured enthalpy and conformational entropy to reveal Fat10-induced sites of local disorder in the substrate. Our experimental and computational data suggest that nonspecific collisions with the proteasome-targeting tag destabilize the substrate. These results highlight that the proteasome-targeting tags's plasticity regulates proteasomal degradation.

## Results

### Fat10 undergoes rapid proteasomal degradation

Although UBLs are structurally similar, they are diverse in sequence landscape. The N-terminal and C-terminal Fat10 domains (Fat10D1 and Fat10D2) share 29% and 36% of their sequence identity with ubiquitin, respectively (*Figure 1A–C*). Moreover, the two domains of Fat10 share only 18% of their

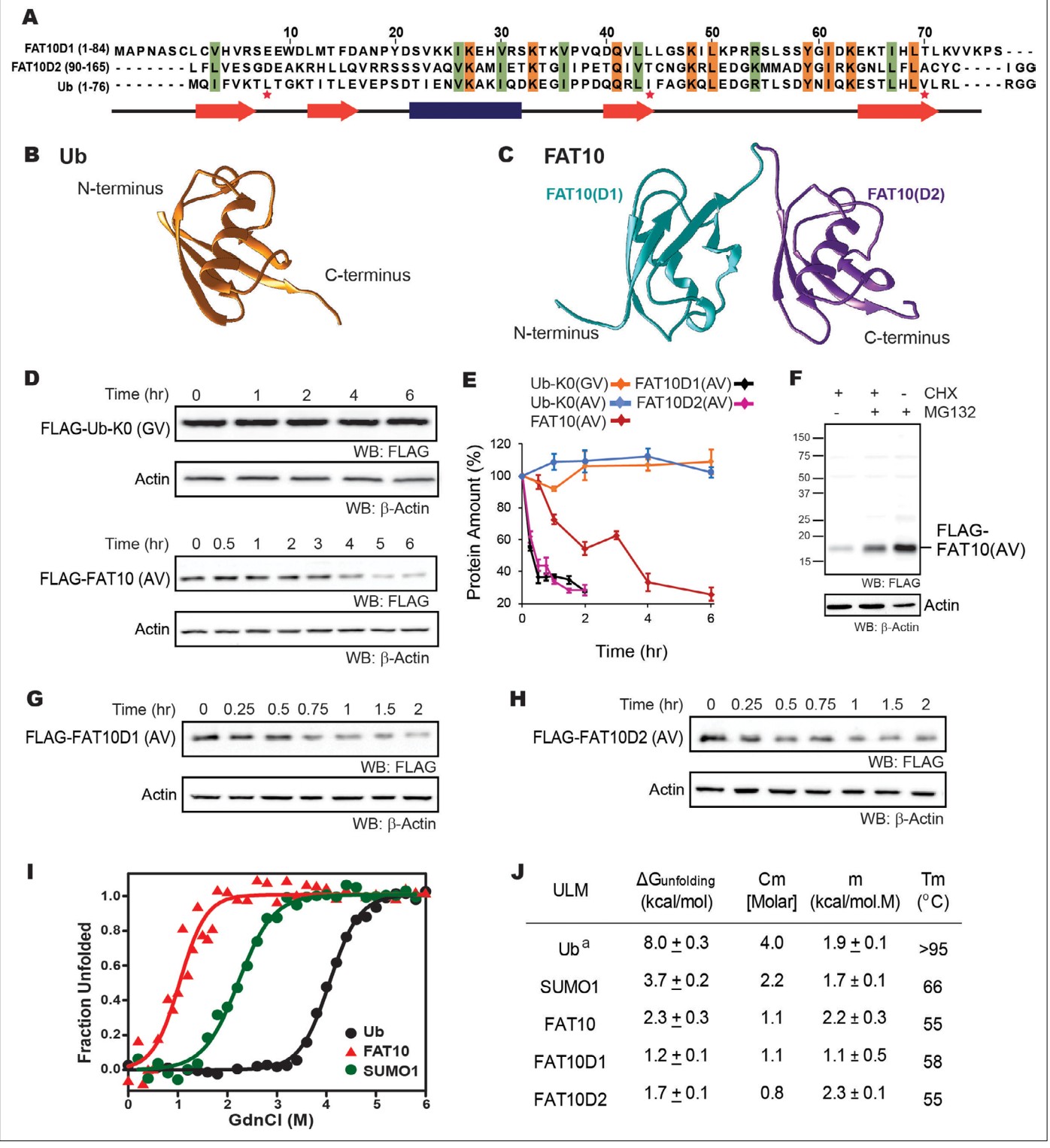

**Figure 1.** The in-cellulo and in-vitro stability of Fat10 and its domains were compared against ubiquitin. (**A**) Structure-based sequence alignment of Fat10D1 (PDB: 6gf1), Fat10D2 (PDB: 6gf2), and ubiquitin (PDB: 1ubq). Conserved hydrophobic and identical residues in Fat10D1, Fat10D2, and ubiquitin are highlighted in light green and orange colors. The L8-I44-V70 residues that create a 'hot spot' of interactions in ubiquitin are marked with a red asterisk. (**B**) Structure of ubiquitin (1UBQ; orange) and (**C**) Homology model structure of full-length Fat10 where Fat10D1 is colored cyan, and Fat10D2 is colored purple. (**D**) FLAG-UbK0(GV) and FLAG-Fat10(AV) protein levels are plotted against time. The C-terminal GG residues are substituted with GV or AV to prevent conjugation to the cellular substrates. HEK293T cells were transfected with either FLAG-Ub or FLAG-Fat10, treated with Cycloheximide, and lysed at different time points. The lysates were separated on SDS PAGE gels and blotted with anti-FLAG antibodies. (**E**) Quantified protein levels

*Figure 1 continued on next page*

*Figure 1 continued*

of FLAG-UbK0(GV) (n=3), FLAG-UbK0(AV) (n=2), and FLAG-Fat10(AV) (n=3) are plotted against time after normalizing with β-Actin. (**F**) HEK293T cells were transfected with Fat10, treated with/without Cycloheximide, and proteasomal inhibitor MG132. The lysates were separated on SDS PAGE gels and blotted with anti-FLAG antibodies. (**G**) Similar to (**D**), showing degradation of FLAG-Fat10D1(AV) and (**H**) FLAG-Fat10D2(AV) after cycloheximide treatment. Quantified protein levels (n=3) of FLAG-Fat10D1(AV) and FLAG-Fat10D2(AV) are plotted in (**E**). (**I**) GdnCl melt curves of Fat10, Ub, and SUMO1. Normalized mean ellipticity shift is plotted against GdnCl concentration. (**J**) A table with the stability parameters of Ub, SUMO1, Fat10, and Fat10 domains is provided. (a: Reference *Wintrode et al., 1994*).

The online version of this article includes the following source data and figure supplement(s) for figure 1:

**Source data 1.** Original file for the Western blot analysis in *Figure 1D–H* (anti-FLAG, anti-actin).

**Figure supplement 1.** Comparison of Fat10 with other ULMs.

**Figure supplement 1—source data 1.** Original file for the western blot analysis in *Figure 1—figure supplement 1F and G* (anti-FLAG).

**Figure supplement 1—source data 2.** Original file for the western blot analysis in *Figure 1—figure supplement 1E* (anti-FLAG, anti-actin).

**Figure supplement 2.** Characterizing purified Fat10.

**Figure supplement 3.** Characterizing individual D1 and D2 domains of Fat10.

sequence identity. Important regions like the L8-I44-V70 hydrophobic patch, an interaction hotspot in ubiquitin, are absent in Fat10. A phylogenetic tree analysis based on available ULM structures showed that among the two Fat10 domains, the C-terminal Fat10D2 is structurally closer to ubiquitin (*Figure 1—figure supplement 1A–D*). Given the poor sequence identity between ubiquitin and Fat10, their cellular levels could be distinctly regulated.

The degradation rate of ubiquitin and Fat10 was compared in cells using cycloheximide chase assay. Non-conjugable variants UbK0-GV, UbK0-AV and Fat10-AV were expressed in HEK293T3 cells, and the protein levels were measured against time (*Figure 1D*, *Figure 1—figure supplement 1E*). Fat10 levels dropped sharply compared to ubiquitin, suggesting that Fat10 undergoes rapid degradation (*Figure 1E*). Treatment with a proteasomal inhibitor restores Fat10 (*Figure 1F*), indicating that Fat10 undergoes proteasomal degradation. The protein levels of individual Fat10 domains were also studied. Both domains degraded faster than the full-length Fat10 (*Figure 1E, G and H*), suggesting that the Fat10 is more stable than the individual domains. The isolated Fat10 domains were also degraded via the proteasomal pathway (*Figure 1—figure supplement 1F and G*). Protein cofactors specific to the full-length Fat10 may bind and bury its partially disordered regions to reduce proteasomal degradation. Transient interactions between the isolated domains may also bury the disordered region and provide a more compact structure to the full-length Fat10, reducing its degradation.

## Thermodynamic characterization of Fat10

To study the correlation between Fat10 degradation and its structural and thermodynamic properties, Fat10 unfolding was studied in a purified system. Fat10 was recombinantly expressed and purified. The far-UV Circular Dichroism (CD) spectra and two-dimensional $^{15}$N Heteronuclear Single Quantum Coherence ($^{15}$N-HSQC) NMR spectra of Fat10 suggested a well-folded tertiary structure (*Figure 1—figure supplement 2A, B*). The thermodynamics of unfolding Fat10 and other ULMs were measured by chemical denaturation using guanidinium hydrochloride (GdnHCl). Gibbs free energy of unfolding ($\Delta G_{unfolding}$) for ubiquitin was 8 kcal/mol as shown in *Figure 1I–J*. $\Delta G_{unfolding}$ for another ULM, SUMO1 was lower than ubiquitin (3.7 kcal/mol). However, $\Delta G_{unfolding}$ was lowest for Fat10 (2.3 kcal/mol). Ubiquitin also has high thermodynamic stability, and its melting temperature (Tm) is greater than 95 °C (*Wintrode et al., 1994*), whereas Fat10 unfolded with a Tm of 55 °C (*Figure 1—figure supplement 2C*), suggesting Fat10 has a malleable structure compared to ubiquitin. Differential scanning fluorimetry measurements reported similar differences in Tm between Fat10 and ubiquitin (*Aichem et al., 2018*).

Transient interactions between the Fat10 domains may bury partially disordered regions and increase the thermodynamic stability of the full-length protein. In that case, the unfolding energies of individual domains shall be lower than the full-length protein (*Batey et al., 2008*). The two Fat10 domains were isolated, and their energetics were measured. Far-UV CD and $^{15}$N-HSQC spectra confirmed that the purified isolated Fat10 domains are folded (*Figure 1—figure supplement 3A–C*). GdnHCl-induced and temperature-induced denaturation of the Fat10 domains were carried out (*Figure 1—figure supplement 3D–F*). $\Delta G_{unfolding}$ of individual domains is lower than full-length Fat10

(*Figure 1J*), suggesting transient interactions between the Fat10 domains stabilize the full-length protein. $\Delta G_{unfolding}$ and Tm values indicate that Fat10 has substantially lower thermodynamic stability than ubiquitin.

## Fat10 has low resistance to mechanical unfolding

Protein mechanical unfolding by molecular motors, such as the proteasomal ATPases, can be simulated by steered molecular dynamics (*Lu et al., 1998*). The mechanical unfolding of Fat10 was studied in the explicit solvent by Adaptive Steered Molecular Dynamics (ASMD) (*Figure 2A*, *Figure 2—video 1*, *Figure 2—figure supplement 1*, *Mücksch and Urbassek, 2016*). A linear di-ubiquitin ($Ub^2$), whose size is similar to Fat10, was used for comparison. While the work required to unfold and linearize $Ub_2$ was 735 kcal/mol, the same for Fat10 was 581 kcal/mol (*Figure 2B*). The lower work required to unfold Fat10 corroborated the lower value of $\Delta G_{unfolding}$ measured during Fat10 denaturation (*Figure 1J*). The simulation was repeated for each isolated Fat10 domain and compared to monoubiquitin. The work required to unfold the isolated Fat10 domains was lower than ubiquitin by 30–60 kcal/mol (*Figure 2B*). Resistance during mechanical unfolding arises from cooperative packing interactions between buried sidechain atoms (*Brockwell et al., 2005*). The lower resistance in Fat10 suggests weaker cooperative packing interactions at the protein's buried core.

Weak cooperative intramolecular interactions should result in faster unfolding kinetics. Fat10 unfolding kinetics was studied by MD simulations at high temperatures. Three µs (300ns * 10 replica) simulations were performed at 450 K for $Ub_2$ and Fat10 (*Figure 3—video 1*). Native intramolecular contacts, including Van der Waals (VdW) interactions and hydrogen bonds (hbonds), decrease over time as the protein unfolds at high temperatures. While $Ub_2$ unfolded gradually in 300 ns, Fat10 unfolded within 40 ns (*Figure 2C*). The native contacts and hbonds in the individual isolated Fat10 domains also disrupted faster than ubiquitin (*Figure 2D*). The ULM proteins adopt a β-grasp fold consisting of a five-strand β-sheet. The β-sheet hbonds were disrupted promptly in Fat10 as opposed to ubiquitin (*Figure 2E and F*), confirming weak cooperative interactions in Fat10.

The fraction of native β-sheet hbonds in Fat10 and ubiquitin were plotted in *Figure 2G–I*. Distinct sequences of inter β-strand hbonds disruption between the Fat10 domains and ubiquitin suggest distinct unfolding pathways. A comprehensive analysis of the secondary structure hbonds during unfolding showed that the $3^{10}$ helix α2 unfolds first in ubiquitin, followed by β3-β4, α1, β3-β5, and β1-β5 contacts (*Figure 2J*). The β1-β2 contacts break at the last step. The unfolding pathway is similar to that observed in equilibrium simulations of ubiquitin (*Piana et al., 2013*). In contrast, the first unfolding event in Fat10D1 is the disruption of β3-β4 contacts, followed by α2, α1, and β1-β5 contacts (*Figure 2K*). The β3-β5 and β1-β2 contacts are disrupted simultaneously at the last step. Similarly, the unfolding pathway of the second domain in Fat10 is distinct from ubiquitin (*Figure 2L*). Altogether, the resistance to mechanical unfolding is lower in Fat10 than in ubiquitin, the kinetics of Fat10 unfolding is faster, and the Fat10 unfolding pathway is distinct from the ubiquitin unfolding pathway.

## The absence of key interactions creates partially unstructured regions in Fat10

The flexible Fat10 structure may sample higher energy states with partially unstructured regions, which was investigated by MD simulations at room temperature. Root Mean Square Fluctuations (RMSF) of Fat10 domains are high at the β1-β2 loop, the C-terminal end of α1, and the α2 loop, indicating regions with high flexibility (*Figure 3—figure supplement 1A*). 2D free energy of Fat10 conformations was calculated as a function of the negative logarithm of root mean square deviation (RMSD) and the radius of gyration (Rgyr) populations. Ubiquitin was used as a control in these simulations. The 2D free energy plot of ubiquitin reflected a single highly populated state (*Figure 3A*), whose representative structure superimposes with the native ubiquitin x-ray structure (*Figure 3B*). In contrast, the native states of Fat10 domains are in equilibrium with higher energy partially unfolded states (*Figure 3A*). Local interactions between β1-β2 and helix α1 are disrupted, and loops at the N- and C-termini of α1 are disordered in these partially unfolded states.

The radius of gyration (Rog) reflects the nature and extent of packing interactions in the protein. Although ubiquitin and the Fat10 domains are similar in size, the Rog of the Fat10 ground state is higher than ubiquitin, suggesting more flexibility and lower packing (*Figure 3C*). VdW and electrostatic interaction energies were calculated between the interacting secondary structures in Fat10 and

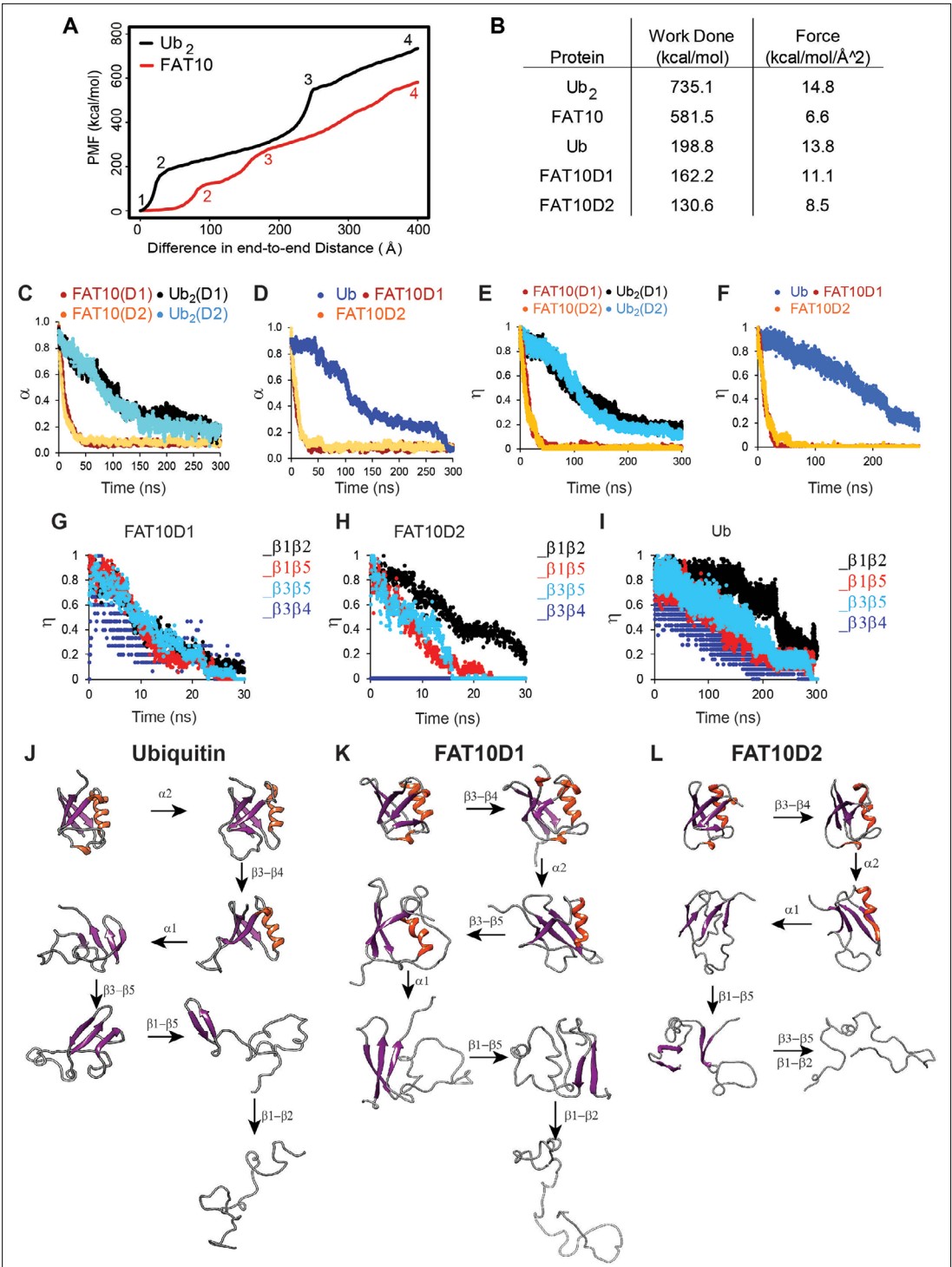

**Figure 2.** Unfolding studies of Fat10, di-ubiquitin (Ub₂), Ub, and individual domains D1/D2 in Fat10 by MD simulations. (**A**) ASMD of Ub₂ and Fat10. The potential Mean Force (PMF) is plotted against the normalized end-to-end distance. Each unfolding event is marked by a number, whose corresponding conformation is given in *Figure 2—figure supplement 1*. (**B**) The work done to unfold the proteins by ASMD is provided. (**C**) Simulations of Fat10, di-ubiquitin (Ub₂), Ub, and individual domains D1/D2 in Fat10 were performed at 450 K. The fraction of native contacts is defined as ($\alpha$) and plotted against time. Fat10(D1) and Fat10(D2) are the D1 and D2 domains in Fat10. Ub₂(D1) and Ub₂(D2) are the two Ub domains in diubiquitin. The data is averaged over ten replicas. (**D**) is the same as (**C**) measured for individual domains Fat10D1, Fat10D2, and Ub. (**E**) The fraction of native beta-sheet backbone hydrogen bonds is defined as ($\eta$) and plotted for Fat10 and Ub₂ against time. (**F**) is the same as (**E**) measured for individual domains Fat10D1, Fat10D2, and Ub. $\eta$ is plotted for β1 to β5 of (**G**) Fat10D1, (**H**) Fat10D2, and (**I**) ubiquitin. The intermediate structures of the unfolding pathway in (**J**) Ub, (**K**) Fat10D1, and (**L**) Fat10D2 are shown, which were inferred from the fraction of backbone hbonds from ten replicas in the MD simulations.

*Figure 2 continued on next page*

*Figure 2 continued*

The online version of this article includes the following video and figure supplement(s) for figure 2:

**Figure supplement 1.** Studying the mechanical unfolding of Fat10 and ubiquitin by steered MD.

**Figure 2—video 1.** Mechanical unfolding of Fat10 and di-ubiquitin by MD simulations is shown here.

https://elifesciences.org/articles/91122/figures#fig2video1

ubiquitin to study the basis of reduced packing interactions. The VdW interaction energy was comparable between Fat10 domains and ubiquitin (*Figure 3D*). The electrostatic interaction energy was also similar in all regions except between β2 and α1, where Fat10 has a significantly lower negative value than ubiquitin, suggesting fewer electrostatic contacts (*Figure 3D*). Two hydrogen bonds and a salt bridge between β2 and α1 are present in ubiquitin but absent in Fat10, which reduces the interaction between β2 and α1 (*Figure 3—figure supplement 1B*). Another salt bridge between the α2 loop and α1 is also exclusive to ubiquitin (*Figure 3—figure supplement 1C*). These salt bridges may be crucial to the stability and compactness of ubiquitin, and their absence makes Fat10 flexible.

When these salt bridges were disrupted in ubiquitin by substitution, the free energy landscape changed, giving rise to partially unfolded forms similar to Fat10 (*Figure 3—figure supplement 1D*). The RMSF values of mutant ubiquitin increase at the C-terminal end of α1 and the β4-α2 loop. Principle component analysis (PCA) and the RMSF values suggested that the ubiquitin mutants have increased dynamics in the regions of broken salt bridges (*Figure 3—figure supplement 1E-G*). When the unfolding kinetics between mutants and wt protein were compared by simulations at 450 K, the mutants unfolded much faster than ubiquitin but were similar to Fat10 domains (*Figure 3E*). The thermodynamic stability of ubiquitin mutants was measured experimentally. Disrupting the salt bridges reduces the melting point of ubiquitin by 50 °C, suggesting the importance of the salt bridges for ubiquitin stability (*Figure 3F-G*). The salt bridges were engineered in the N-terminal Fat10 domain by appropriate substitutions. The protein with engineered salt bridges populated the ground state but not the partially unfolded states (*Figure 3—figure supplement 2*). RMSF and PCA analysis suggested that engineered salt bridges reduced the fluctuations and increased rigidity in the domain. Overall, critical electrostatic interactions present between α1 and β1β2 and between α1 and α2 loop stabilize the ubiquitin fold. These interactions are absent in Fat10, creating a flexible structure in dynamic equilibrium with partially unfolded forms.

## Weak hydrogen bonds and high conformational entropy detected in Fat10

Protein thermodynamic stability depends on the energy of enthalpic interactions and conformational entropy. The temperature dependence of backbone amide proton chemical shifts is an excellent reporter of backbone hbond strength, which contributes to enthalpic interactions (*Doyle et al., 2016*). The temperature dependence of the amide proton chemical shifts is the temperature coefficient Tc, where $Tc = \Delta\delta^{NH}/\Delta T$ (*Cierpicki and Otlewski, 2001*). Higher negative Tc values suggest weaker hbonds and disorder propensity, while lower negative Tc values suggest stronger hbonds and structural rigidity. Tc's were measured in Fat10 and plotted against each residue in *Figure 4A*, and the fitting errors are provided in *Figure 4—figure supplement 1A*. In the N-terminal Fat10 domain, residues in the region β1 to β2 and the loop between including α2 had high negative Tc values, suggesting weaker hbonds and disorder propensity in these regions (*Figure 4A*). Similarly, several residues in the β1, β2, α1, and α2 loop in the C-terminal domain had high negative Tc values.

Tc's were measured in ubiquitin and plotted against each residue in *Figure 4B* (fitting errors in *Figure 4—figure supplement 1B*). The loop between β1 and β2 had high negative Tc values in ubiquitin, which correlates with the lack of interactions and disorder propensity in this region (*Peters and de Groot, 2012*). Due to the low sequence similarity between Fat10 domains and ubiquitin, comparing the Tc values between individual residues of the two proteins is challenging. Instead, Tc values were averaged over the different protein segments and compared (*Figure 4—figure supplement 1C*). The difference in averaged Tc values between the N-terminal domain and ubiquitin indicates that Fat10 hbonds are weaker in the β1β2 region and the α2 loop (*Figure 4C and D*). Similarly, the β1β2 regions and α1 have weaker hbonds in the C-terminal domain. The Tc measurements of the isolated Fat10 domains yielded similar results (*Figure 4—figure supplement 1D-J*).

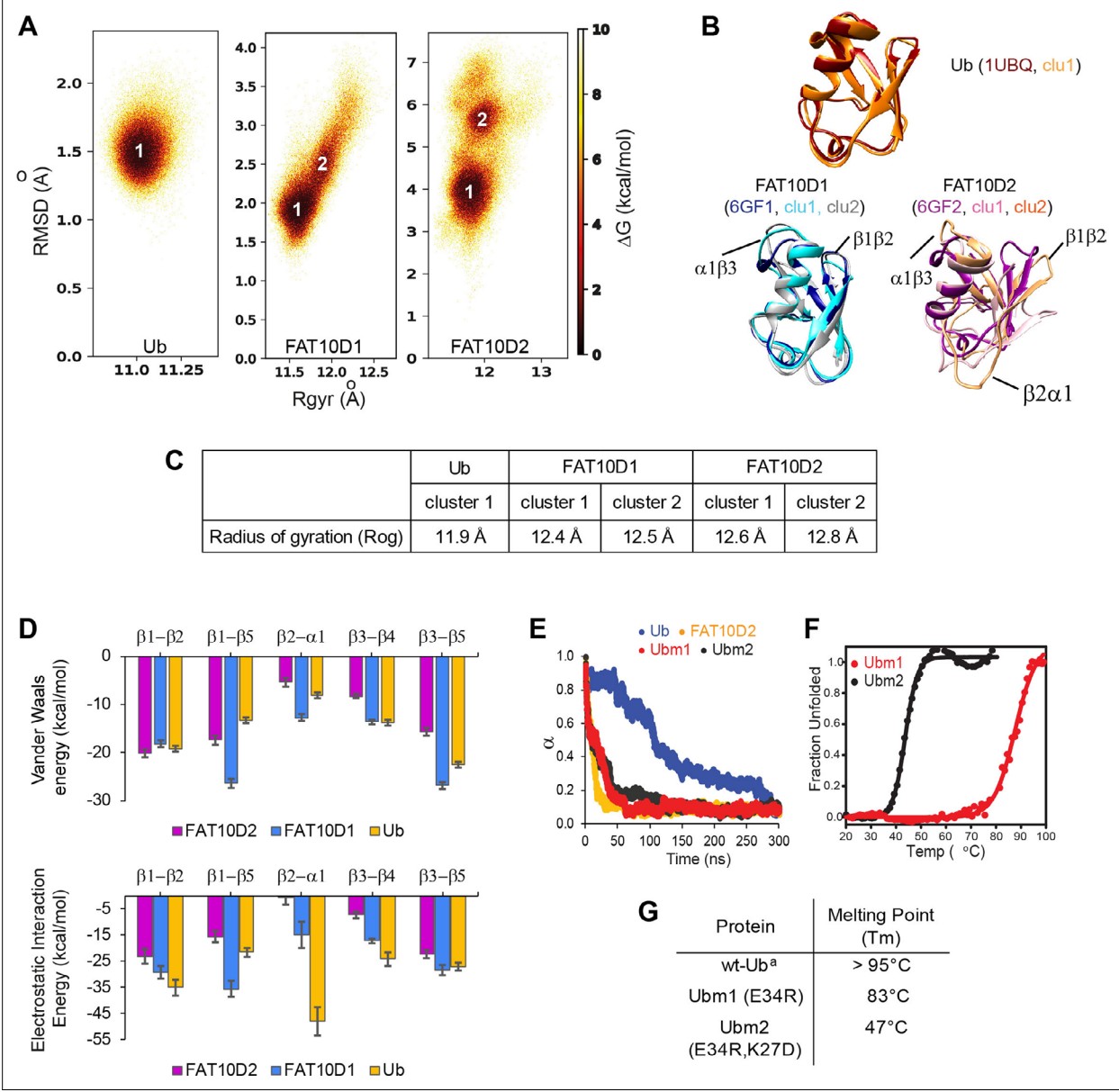

**Figure 3.** A comparison of Ub and Fat10 energetics was investigated by all-atom MD simulations. (**A**) The free energy landscape of Fat10 domains and ubiquitin are plotted as a function of RMSD and radius of gyration (Rgyr) obtained from simulations across three replicas (3x2.5 μs) performed at 300 K. The minima from each cluster are numbered. (**B**) The corresponding conformation of each cluster in (**A**) is shown. The structures of the proteins, denoted by their pdb IDs, are provided for comparison. (**C**) The radius of gyration of the entire protein (Rog) for the minimas in Ub, Fat10D1, and Fat10D2 are provided. The Rgyr values do not account for long loops in the protein, while the Rog values include the complete protein. (**D**) The mean Van der Waals energy of interactions and electrostatic energy of interactions between different pairs of secondary structures obtained from simulations is plotted for Fat10 domains and ubiquitin. The error bars denote the standard deviation. One-way annova was performed (B1-B2; $P = 0.0892$; $P>0.05$, B1-B5; $P = 0.0060$; $P<0.05$, B2-A1; $P = 0.0019$; $P<0.05$, B3-B4; $P = 0.0028$; $P<0.05$, B3-B5; $P = 0.1196$ ; $P>0.05$) (**E**) The fraction of native contacts in Ubm1, Ubm2, Ub, and Fat10 domain against time at 450 K MD simulations. (**F**) The thermal melt curve of Ubm1 and Ubm2, where the change in ellipticity is normalized and plotted against temperature. (**G**) The melting point of ubiquitin mutants. (a: Reference *Wintrode et al., 1994*).

The online version of this article includes the following video and figure supplement(s) for figure 3:

**Figure supplement 1.** Effect of salt bridge substitutions on ubiquitin structure.

**Figure supplement 2.** Effect of salt bridge introduction on FAT10D1 structure.

**Figure 3—video 1.** The thermal unfolding of Fat10 domains and ubiquitin is shown here.

https://elifesciences.org/articles/91122/figures#fig3video1

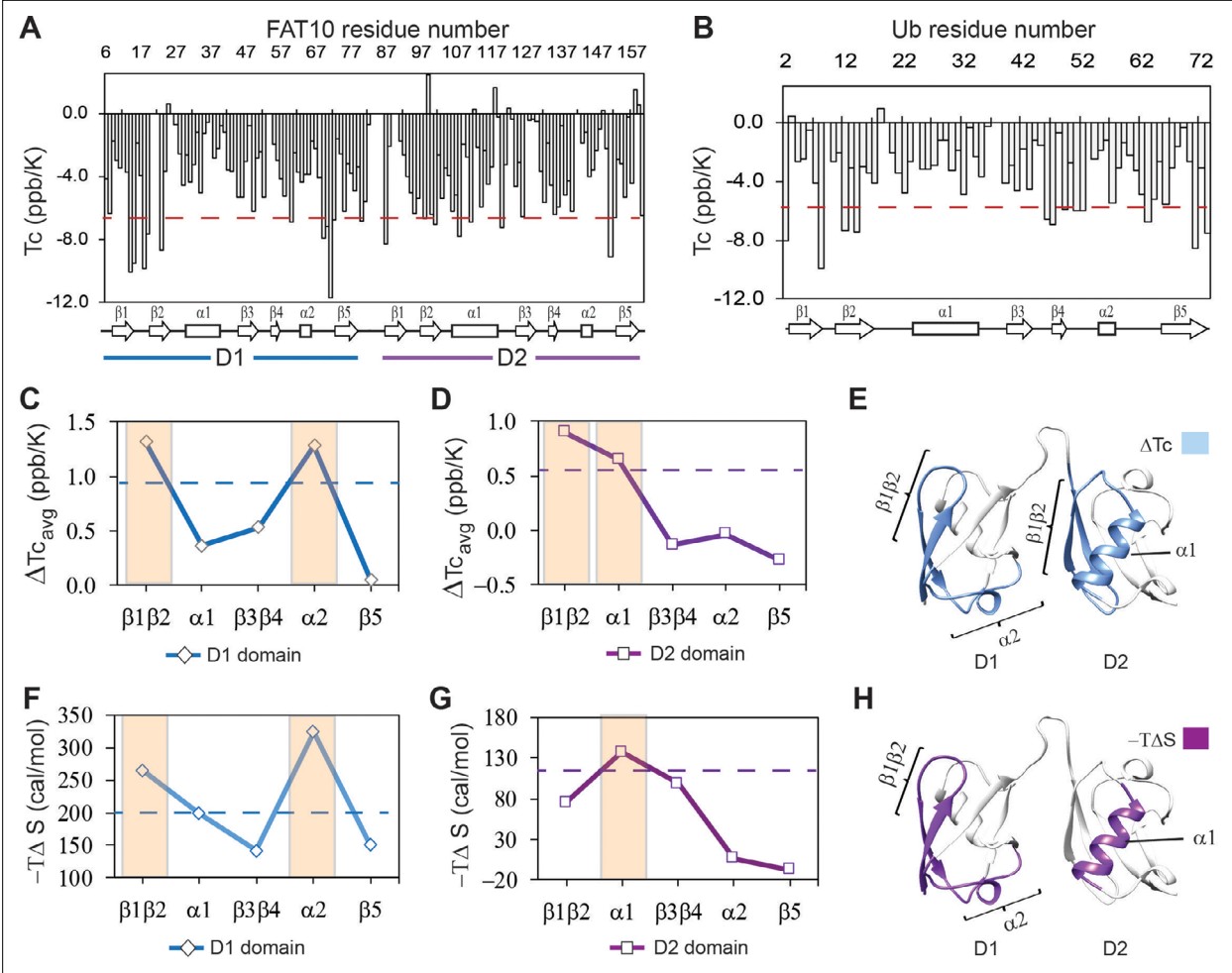

**Figure 4.** The local hbond stability and conformational entropy in Fat10 were measured by NMR spectroscopy. Temperature coefficients (Tc) are plotted for Fat10 and ubiquitin in (**A**) and (**B**), respectively. The horizontal red line is (mean – S.D.), where the mean value is negative. High negative Tc values suggest weaker hbonds and disorder propensity. (**C**) The difference in average temperature coefficients ($\Delta Tc_{avg}$) between the N-terminal Fat10 domain (**D1**) and ubiquitin, where $\Delta Tc_{avg} = Tc(Ub)_{avg} - Tc(D1)_{avg}$. The blue dashed line is mean + error. (**D**) The difference in averaged temperature coefficients ($\Delta Tc_{avg}$) between the C-terminal domain and Ub, where $\Delta Tc_{avg} = Tc(Ub)_{avg} - Tc(D2)_{avg}$. The purple dashed line is the mean + error. Higher $\Delta Tc_{avg}$ values suggest weaker hbonds and destabilization in these Fat10 regions than ubiquitin. (**E**) The segments with high $\Delta Tc_{avg}$ values are colored light blue on the Fat10 structure. (**F**) The difference in conformational entropy $-T\Delta S_{conf}$, where $\Delta S_{conf} = S_{conf}^{Ub} - S_{conf}^{D1}$, was averaged for the various segments and plotted. The entropy values were calculated from the order parameters measured in *Figure 4—figure supplement 2*. The broken line denotes (mean + error). (**G**) Same as (**F**) except conformational entropy is calculated for the C-terminal Fat10 domain, such that $\Delta S_{conf} = S_{conf}^{Ub} - S_{conf}^{D2}$. (**H**) The segments with $-T\Delta S$ more than (mean + SD) are colored purple on Fat10 domains. Higher values of $-T\Delta S$ suggest increased conformational flexibility in these Fat10 regions compared to ubiquitin.

The online version of this article includes the following figure supplement(s) for figure 4:

**Figure supplement 1.** The temperature dependence of amide chemical shifts.

**Figure supplement 2.** Measurement of backbone dynamics in Ub and Fat10.

To estimate the conformational entropy in Fat10, the backbone order parameters ($S^2$) were measured by standard NMR relaxation experiments. The NMR spin-lattice relaxation parameter R1, spin-spin relaxation parameter R2, and heteronuclear NOEs (hetNOE) were measured in Fat10 and ubiquitin (*Figure 4—figure supplement 2*). $S^2$ values were calculated using the Lipari-Szabo Model-Free Analysis method and averaged over the protein segments. The difference in averaged $S^2$ between Fat10 and ubiquitin was converted to conformational entropy $-T\Delta S$ (*Sharp et al., 2015*; *Zeng et al., 2017*) and plotted in *Figure 4F–H*. The higher value of $-T\Delta S$ suggests higher dynamics and flexibility. The β1β2 and α2 loops had higher conformational entropy in the first Fat10 domain than ubiquitin (*Figure 4F and G*). In the second domain, β1β2, β3β4, and helix α1 were more entropic compared to

ubiquitin, the most significant being α1 (*Figure 4G and H*). Overall, salt bridges increase interactions between α1 and β1β2 and between α1 and the α2 loop in ubiquitin. In their absence, these regions in Fat10 have weaker hbonds and higher conformational entropy.

## Fat10 increases substrate unfolding and degradation in cells

We then studied the degradation of Fat10 conjugated substrates in cells by a cycloheximide assay using GFP as the model. Fat10-conjugated GFP degraded by 60% within 4 hr after cycloheximide treatment, much faster than diubiquitin-conjugated GFP (*Figure 5A*, *Figure 5—figure supplement 1A*). To study the effect of individual Fat10 domains, we compared the degradation of GFP in Fat10D1-GFP and Ub-GFP (*Figure 5B*, *Figure 5—figure supplement 1B*). The Fat10D1-GFP degraded by 40% within 4 hr, whereas the levels of Ub-GFP remained unchanged. It is unclear if the higher degradation rate is exclusive because the proteasome-targeting tag Fat10 degrades rapidly, thereby accelerating substrate degradation, or if it also affects the substrate's thermodynamic stability and induces partially unstructured regions. To study changes in the substrate thermodynamic stability due to Fat10 in cellular conditions, a heterologous system has to be used where Fat10 substrates are not degraded. The bacterial Pup-proteasome system functions by a distinct mechanism (*Striebel et al., 2014*) that does not recognize or degrade the eukaryotic Ub/Fat10 conjugated proteins and can be used as a heterologous host. The CRABP1 protein was chosen as the model substrate. CRABP1 can be engineered to bind a fluorescent dye that is quenched in the native state but not in the unfolded state (*Ignatova and Gierasch, 2004*; *Figure 5C*). Cells expressing CRABP1 were treated with various urea concentrations. The difference in fluorescent signals between 0 M and 3 M Urea was lowest in apo CRABP1, higher in Ub-CRABP1, and highest in Fat10-CRABP1, suggesting the CRABP1 stability is lowest when conjugated to Fat10 (*Figure 5—figure supplement 1C-E*). $\Delta G_{unfolding}$ was 5.9 kcal/mole for CRABP1, 4.8 kcal/mole for Ub-CRABP1, and 2.4 kcal/mole for Fat10-CRABP1 (*Figure 5D–E*, *Figure 5—figure supplement 1F*). The difference of 2.4 kcal/mol between Fat10-CRABP1 and Ub-CRABP1 suggested a significant increase in substrate unfolding by Fat10 conjugation.

## Fat10 induces partially unfolded forms in the substrate

In highly crowded cellular environments, proteins experience various physical interactions, such as the exclusion volume effect and nonspecific transient interactions. To study whether the substrate destabilization is solely due to Fat10 and not the above effects, the impact of Fat10 was studied in purified proteins. Producing purified Fat10 isopeptide conjugated substrates is technically challenging owing to the lack of a suitable Fat10 E3-substrate pair active under in vitro conditions. Hence, Fat10 was covalently conjugated at the N-terminus of a model ultra-stable substrate protein like the Cyan Fluorescent Protein (CFP) to mimic the N-terminal isopeptide conjugation (*Ciechanover and Ben-Saadon, 2004*; *Figure 6A*). The free energy of unfolding CFP was calculated by measuring CFP fluorescence during chemical denaturation. CFP is a well-folded substrate with high thermodynamic stability and $\Delta G_{unfolding} = 11$ kcal/mol. The change in unfolding energy of CFP is modest $\Delta\Delta G_{unfolding} = 0.2$ kcal/mol when conjugated with ubiquitin (*Figure 6B*, *Figure 6—figure supplement 1A*). The modest decrease in CFP stability correlates with the finding that ubiquitin conjugation is insufficient for the direct degradation of well-folded substrates and requires Cdc48 (*Olszewski et al., 2019*). Interestingly, the $\Delta\Delta G_{unfolding}$ was 3.0 kcal/mol and 3.8 kcal/mol when CFP is conjugated to Fat10D1 and Fat10D2, respectively (*Figure 6B*, *Figure 6—figure supplement 1B*), suggesting Fat10 domains destabilize the substrate 15-fold more than ubiquitin. Change in free energy could be due to the nonspecific or specific interactions between Fat10 and CFP (*Bigman and Levy, 2020*). Specific interactions should depend on the complementary surface properties of Fat10 and CFP and should differentiate between the full-length Fat10 and the smaller individual domains. CFP's thermodynamic stability after conjugation to the full-length Fat10 reduced by 3kcal/mol, similar to the isolated domains, suggesting that nonspecific interactions due to conjugation reduce CFP stability.

Fat10 may increase the partially unfolded regions in the substrate, which is critical to interact with the proteasome. The susceptibility of partially unfolded conformations to a typical protease thermolysin was measured in CFP, ubiquitin-CFP, and Fat10-CFP (*Figure 6B–D*, *Figure 6—figure supplement 1C-F*). Although CFP was stable in the presence of protease, ubiquitin-CFP and Fat10-CFP were proteolyzed. The Fat10-CFP proteolysis rate ($k_{obs}$) was significantly more than ubiquitin-CFP at a given thermolysin concentration (*Figure 6D*). The change in observed proteolysis rates with

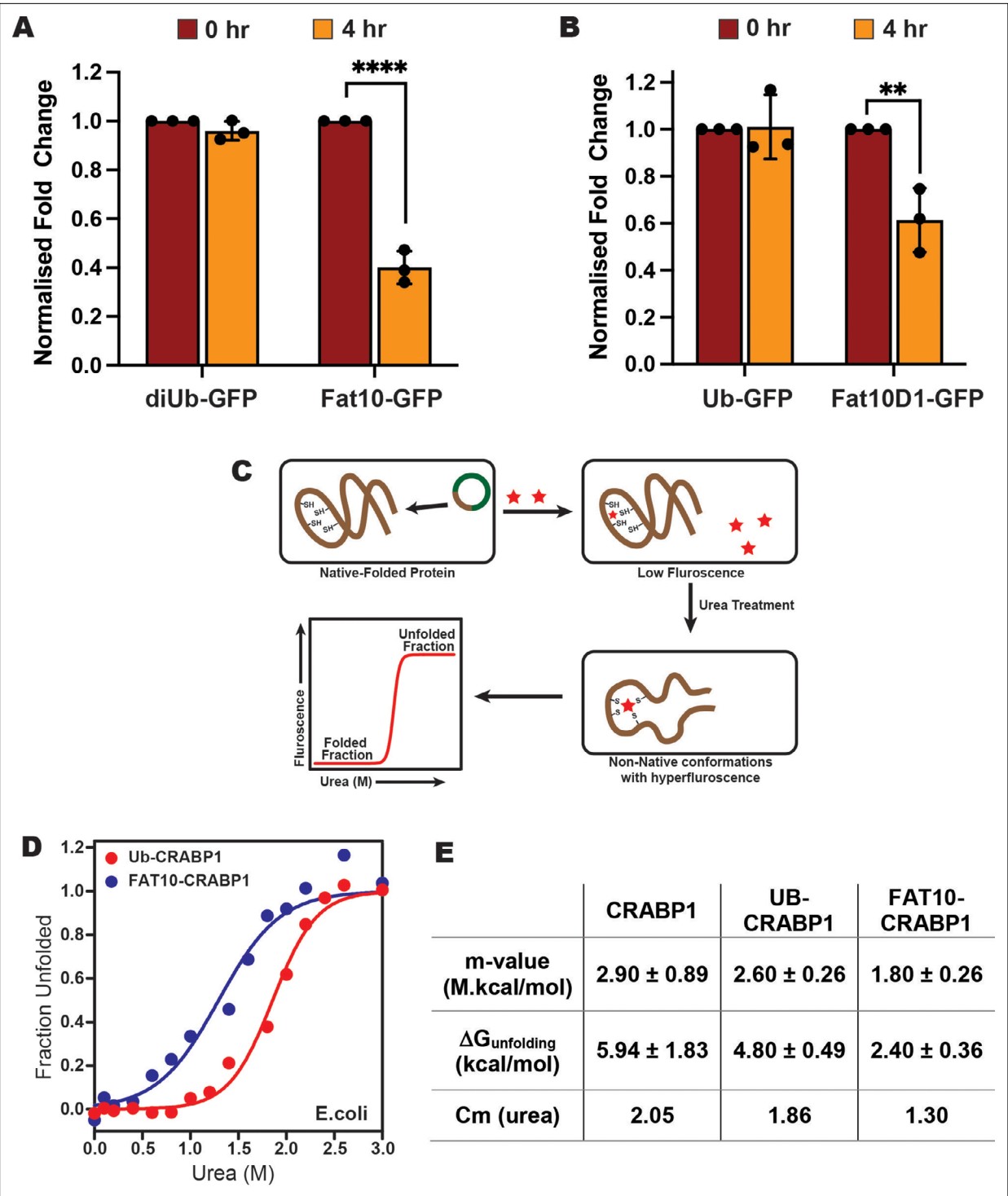

**Figure 5.** Stability comparison of Ub and Fat10 conjugated substrates in cellular conditions. (**A**) Comparison of diUb-GFP and Fat10-GFP levels post 4 hr treatment with Cycloheximide (CHX) in HEK293T cells (n=3, p <0.0001; 2-way ANOVA with Bonferroni's multiple comparison test). (**B**) Comparison of monoUb-GFP and Fat10D1-GFP levels post 4 hr treatment with Cycloheximide (CHX) in HEK293T cells (n=3, p = 0.0023; 2-way ANOVA with Bonferroni's multiple comparison test). (**C**) The schematic for studying the substrate stability in a heterologous cellular system is provided, where the CRABP1 protein, capable of binding FlAsH-EDT2 dye, was expressed in *E. coli* BL21 (DE3) cells as a substrate. The folded protein quenches the dye, while the unfolded protein releases the quenching. The protocol enables the study of protein unfolding under cellular conditions. (**D**) The cells were treated with different urea concentrations, and the dye fluorescence was measured. The fluorescent signals of Ub-CRABP1 and Fat10-CRABP1 were normalized to plot their denaturation curves against urea concentration. (**E**) Thermodynamic parameters of CRABP1 in cellular conditions when covalently bound to ubiquitin and Fat10, respectively.

*Figure 5 continued on next page*

*Figure 5 continued*

The online version of this article includes the following source data and figure supplement(s) for figure 5:

**Source data 1.** Original file for the western blot analysis in *Figure 5A and B*, *Figure 5—figure supplement 1A and B* (anti-FLAG, anti-actin).

**Figure supplement 1.** Substrate degradation and unfolding by FAT10 min cells.

different thermolysin concentrations ($m_{proteolysis}$) was sevenfold higher in Fat10-CFP than in ubiquitin-CFP (*Figure 6D*). The values of $m_{proteolysis}$ can be correlated to the free energy of proteolysis (*Park and Marqusee, 2004*). The free energy of proteolysis decreased from 8.7 kcal/mol for ubiquitin-CFP to 7.5 kcal/mol for Fat10-CFP (*Figure 6B*), indicating that Fat10 substantially increases the partially unstructured regions in CFP. Our results indicate that Fat10 has a more drastic effect than ubiquitin on substrate stability, suggesting that Fat10 substrates may be degraded independently of the unfoldases (*Aichem et al., 2018*). Degradation of Fat10-GFP was monitored in the HEK293T cells in the

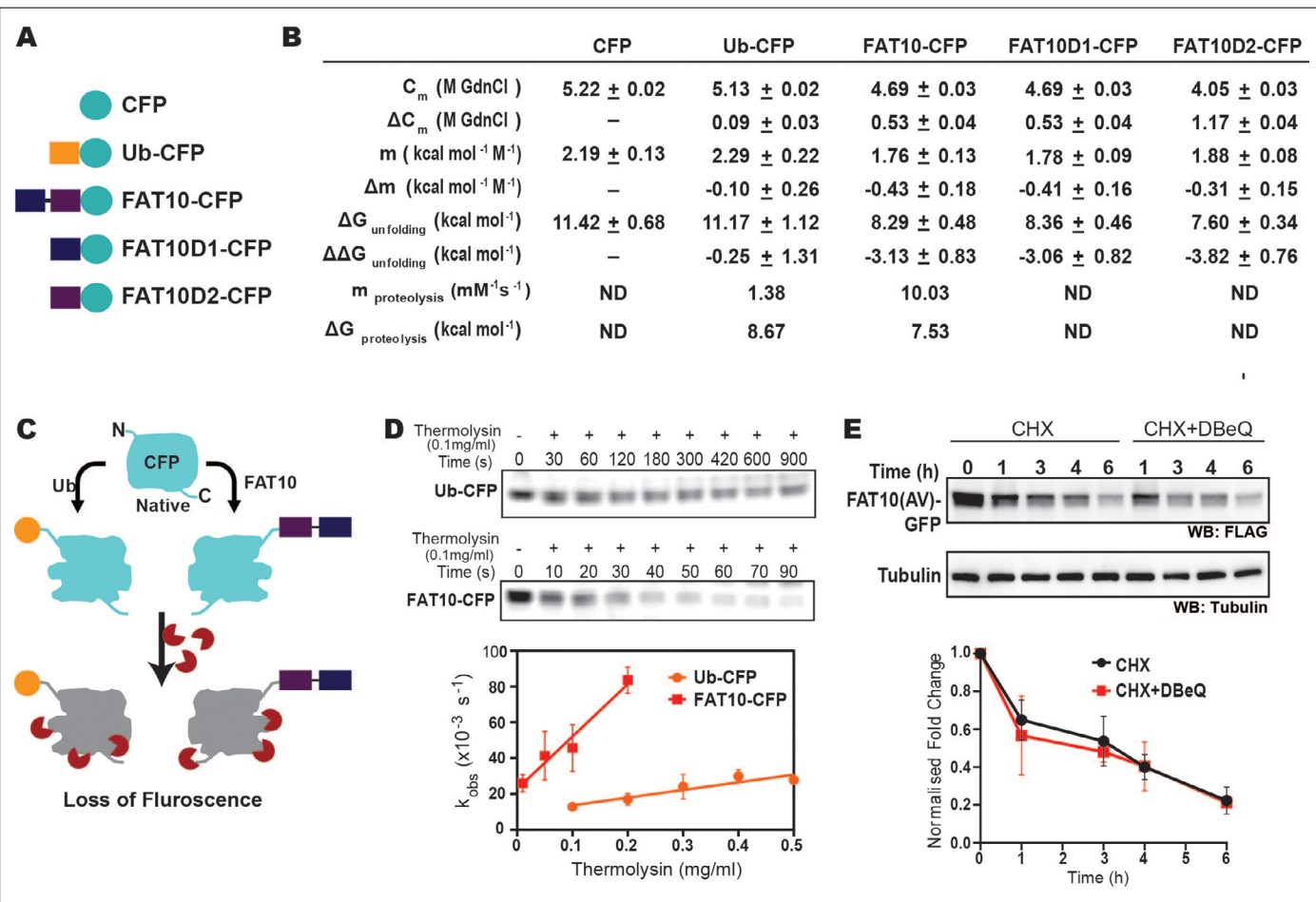

**Figure 6.** The destabilization effect of Fat10 was investigated using CFP as a substrate. (**A**) CFP is fused at the C-terminal end of Ub, Fat10, Fat10D1, and FAT0D2. (**B**) The thermodynamic and proteolysis parameters of CFP in the fusions given in A were studied. These parameters are provided here. (**C**) The schematic of the native state proteolytic cleavage of Ub-CFP and Fat10-CFP is provided. The cleavage reactions were carried out using thermolysin. (**D**) Representative in-gel fluorescence image for native-state proteolysis of Ub-CFP and Fat10-CFP at 0.1 mg ml$^{-1}$ thermolysin is provided. The rate of proteolysis ($k_{obs}$) is plotted for Ub-CFP and Fat10-CFP against different thermolysin concentrations. Error bars denote the standard deviation of the replicates (n=3). (**E**) Degradation of Fat10-GFP in HEK293T cells after cycloheximide treatment without/with p97 inhibitor (DBeQ). Tubulin is used as the loading control. The quantified level of FLAG-Fat10-GFP without/with inhibitor is plotted against time (for n=3).

The online version of this article includes the following source data and figure supplement(s) for figure 6:

**Source data 1.** Original Gels of Ub-CFP and Fat10-CFP corresponding to *Figure 6D* and *Figure 6—figure supplement 1*.

**Source data 2.** Original file for the western blot analysis in *Figure 6E* (anti-FLAG, anti-actin).

**Figure supplement 1.** The effect of ubiquitin, Fat10 and its domains on the thermodynamic stability and proteolysis of CFP.

presence and absence of the Cdc48 inhibitor DBeQ (*Figure 6E*). Fat10-GFP degradation was unperturbed by DBeQ, suggesting that Fat10-modified substrates are degraded independently of Cdc48. Altogether, Fat10 significantly destabilizes substrates and creates partially unstructured regions for direct proteasomal degradation.

## Fat10 reduces local stability in the substrate

To probe the Fat10-induced disorder further, we used a combination of computational and experimental frameworks to detect changes in the substrate's local stability. Ubiquitin was chosen as a well-folded small substrate, as covalent interactions between Fat10 and ubiquitin have been reported previously (*Buchsbaum et al., 2012*). MD simulations studied the impact of Fat10 on the kinetics of substrate ubiquitin unfolding at 450 K. As a control, we used diubiquitin, where the first ubiquitin serves as the tag and the second ubiquitin is the substrate. The substrate ubiquitin in diubiquitin unfolded similarly to its free form (*Figure 7—figure supplement 1A, C*). However, the substrate ubiquitin conjugated to Fat10 domains unfolded faster (*Figure 7—figure supplement 1B, D*), suggesting that Fat10 domains increase the rate of substrate unfolding. Fat10 also enhanced fluctuations in the helix α1 and the loop near α2 in the substrate (*Figure 7A*). It increased the radius of gyration in substrate ubiquitin from 11 Å to 12 Å and created a broad RMSD profile starting from 2 Å to 3 Å, indicating a less compact substrate (*Figure 7B*). The domains had a similar effect and created partially disordered states in the substrate (*Figure 7B*).

Fat10's effect on substrate stability was experimentally measured using $Ub_{45F/W}$ mutant as a substrate, where the intrinsic tryptophan fluorescence reports the thermodynamics of substrate unfolding. $Ub_{45F/W}$ is functionally active, and its thermodynamic stability is similar to ubiquitin (*Khorasanizadeh et al., 1993*). A single tryptophan in the N-terminal Fat10 domain is mutated to Phenylalanine (F) to not interfere with the fluorescence spectra (*Figure 7C*). The free energy of unfolding the substrate reduced from 11.7 kcal/mol to 7 kcal/mol when covalently bound to Fat10 (*Figure 7D*). Individual domains of Fat10 had a similar effect on the substrate (*Figure 7E*). C-terminal conjugation has a higher impact than at the N-termini (*Figure 7F and G*, Ub-Fat10D2 versus Fat10D2-Ub), suggesting that the conjugation site may regulate substrate stability.

To further understand substrate destabilization, we analyzed the tag-substrate and intra-substrate contacts in MD simulations of the Fat10~Ub conjugate. Many tag-substrate contacts were formed between the D2 domain and the β1β2 and β4-α2-β5 regions in the substrate ubiquitin (*Figure 8—figure supplement 1A, B*). We compared the long-range intra-substrate contacts in the free substrate and the Fat10-substrate conjugate (*Figure 8—figure supplement 1C*). The regions where intra-substrate contacts are disrupted overlapped with those where Fat10 contacts the substrate, suggesting that Fat10 interaction disrupts the intra-substrate contacts. To study the higher destabilizing effect of Fat10 compared to ubiquitin, we noted that Fat10 has a higher exposed hydrophobic surface area (23%) than ubiquitin (13%) and can form more hydrophobic interactions with the substrate. Moreover, Fat10 domains are labile and sample partially unfolded forms, exposing their buried hydrophobic residues to enhance hydrophobic interactions with the substrate. We compared the intra-substrate long-range contacts between a linear diubiquitin molecule and Fat10-ubiquitin conjugate. The first ubiquitin serves as the tag in diubiquitin, and the second ubiquitin is the substrate. The number of intra-substrate contacts disrupted in the Fat10-substrate conjugate was higher than diubiquitin, confirming the higher destabilization effect of Fat10 (*Figure 8—figure supplement 2A and B*).

Most intermolecular contacts were observed between the Fat10 D2 domain and the substrate, and few between the D1 domain and substrate. Hence, we conjugated the D2 domain to the N-terminal or C-terminal end of the substrate ubiquitin to study the significance of the conjugation site (*Figure 8A and B*). The C-terminal tail in substrate ubiquitin is dynamic and explores a larger conformational space, providing greater flexibility for the tag to interact with the substrate (*Figure 8B*). Consequently, the intermolecular contacts between the tag and substrate were more evenly distributed (*Figure 8—figure supplement 2C, D*). In addition, a more significant number of the intra-substrate contacts were disrupted (*Figure 8—figure supplement 2A, E, F*), which corroborates the reduced substrate stability upon C-terminal conjugation (*Figure 7F*, Ub-Fat10D2 versus Fat10D2-Ub). These results highlight that conformational dynamics at the conjugation site increase tag-substrate contacts and decrease substrate stability.

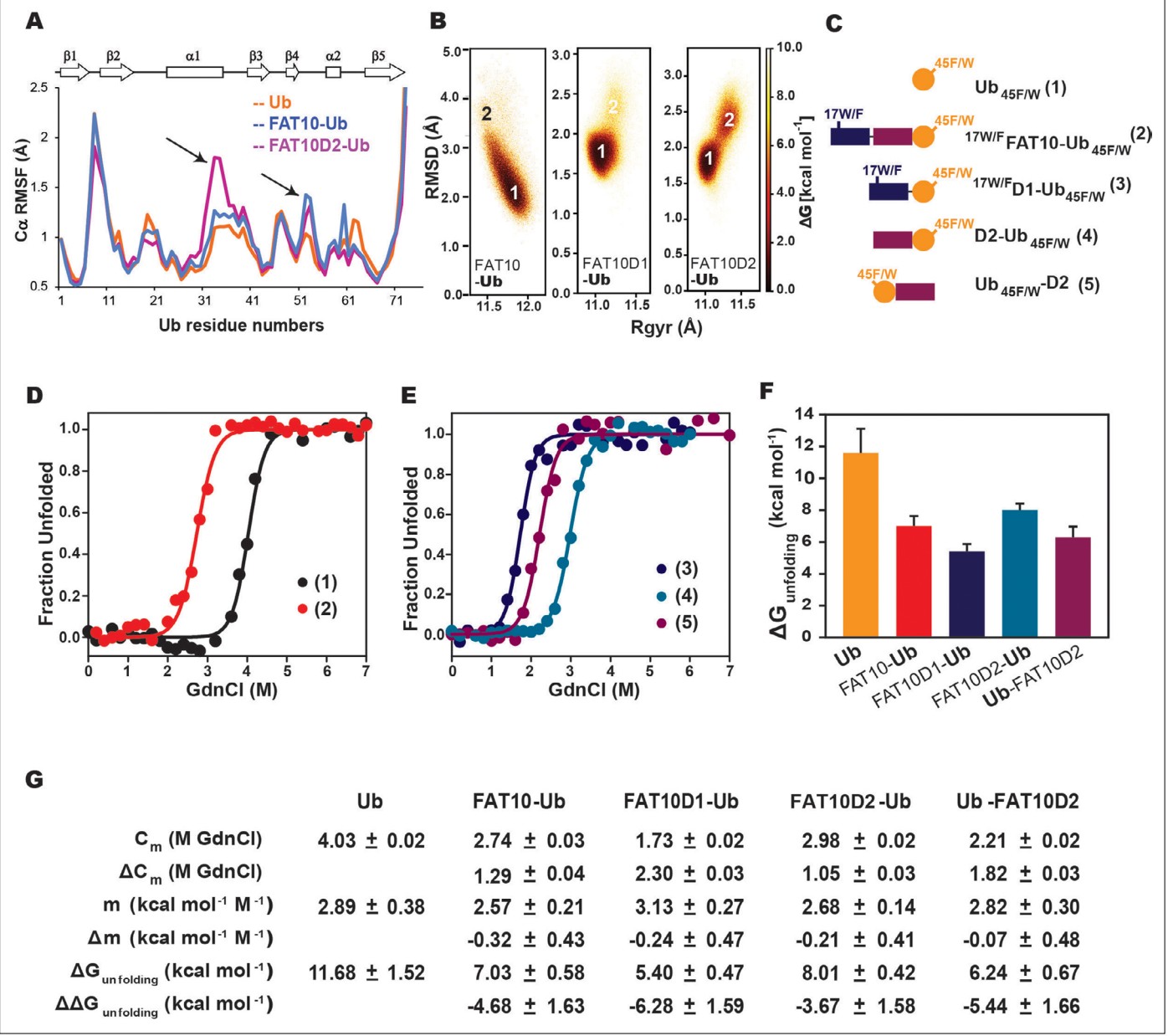

**Figure 7.** Simulations and melting experiments studied the effect of Fat10 on a model substrate ubiquitin. (**A**) Comparing Cα RMSF values of ubiquitin in monoubiquitin, Fat10-Ub, Fat10D1-Ub and Fat10D2-Ub. Black arrows denote regions with higher values in Fat10-conjugated ubiquitin than monoubiquitin. (**B**) The free energy landscape of Ub in Fat10-Ub, Fat10D1-Ub and Fat10D2-Ub are plotted as a function of RMSD and radius of gyration (Rgyr) obtained from simulations across three replicas (3x2.5 μs) performed at 300 K. (**C**) Ubiquitin varieties used in this study where, Ub$_{45F/W}$ is covalently conjugated at the C-terminus of $_{17W/F}$Fat10, $_{17W/F}$Fat10D1, and Fat10D2. Ub$_{45F/W}$ was also fused at the N-terminus of Fat10D2. (**D**) GdnCl melt curves of Ub and Fat10-Ub are shown. The tryptophan fluorescence signals of Ub$_{45F/W}$ were normalized and plotted against the GdnCl concentration. (**E**) GdnCl melt curves were plotted for Fat10D1-Ub, Fat10D2-Ub, and Ub-Fat10D2. (**F**) The free energy of unfolding ubiquitin in free form and covalently bound to Fat10 domains is plotted. (**G**) A table with details of thermodynamic parameters of Ub in free and bound forms is shown.

The online version of this article includes the following figure supplement(s) for figure 7:

**Figure supplement 1.** Molecular dynamics of the substrate ubiquitin when conjugated to the Fat10 domains.

We used NMR spectroscopy to monitor the Fat10-induced changes in the substrate's local stability. Since the destabilizing effect of Fat10 and the D2 domain are similar, and to avoid overlap in the NMR spectra, we studied the conjugates where the D2 domain is conjugated to substrate ubiquitin at its N- and C-termini (Fat10D2-Ub and Ub-Fat10D2). $^{15}$N-edited HSQC spectra of the conjugated proteins showed well-dispersed backbone amide resonances, confirming that they are folded

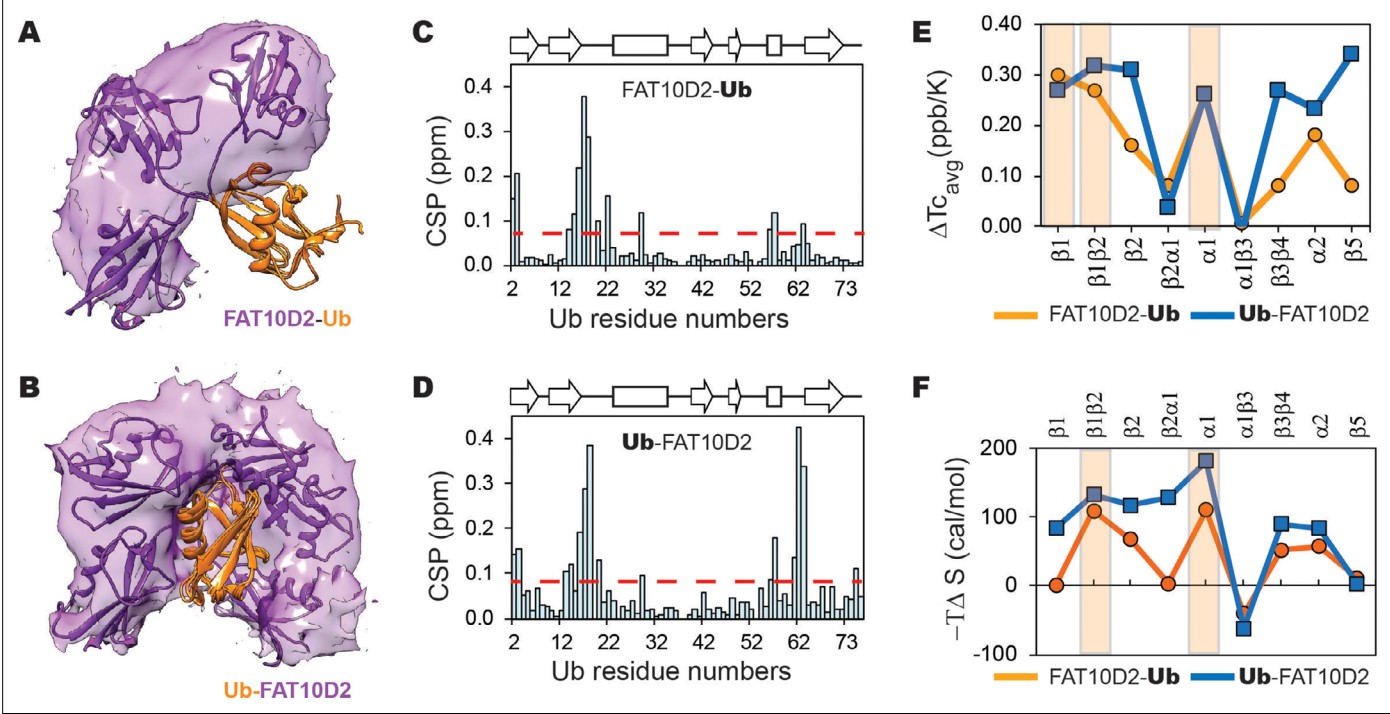

**Figure 8.** The changes in a model substrate ubiquitin, when conjugated to Fat10D2 domain is studied by MD simulations and NMR. The occupancy of the D2 domain around Ub in the simulations is shown as a purple surface for (**A**) Fat10D2-Ub and (**B**) Ub-Fat10D2. A few structures from the simulation are superimposed and shown. The chemical shift perturbations observed in ubiquitin when conjugated to Fat10D2 domain is plotted for (**C**) Fat10D2-Ub and (**D**) Ub-Fat10D2. (**E**) The difference in mean temperature coefficients between Fat10D2-Ub and free Ub is plotted for the various Ubiquitin segments. The orange line is $\Delta Tc_{avg}$ = Tc(Ub) $_{avg}$ - Tc(Fat10D2-Ub) $_{avg}$. The blue line is the same for Ub-Fat10D2, where $\Delta Tc_{avg}$ = Tc(Ub)$_{avg}$ - Tc(Ub-Fat10D2)$_{avg}$. A light orange box highlights the regions with high $\Delta Tc_{avg}$ in Fat10D2-Ub. Ub-Fat10D2 has additional regions with $\Delta Tc_{avg}$ values. (**F**) The difference in conformational entropy -T$\Delta$S of ubiquitin between free Ub and Fat10D2-Ub, where $\Delta S = S^{Ub} - S^{Fat10D2-Ub}$, was averaged for the various secondary structures and loops (orange). The same was plotted for Ub-Fat10D2 in blue. A light orange box highlights the regions with high -T$\Delta$S in Fat10D2-Ub.

The online version of this article includes the following figure supplement(s) for figure 8:

**Figure supplement 1.** Inter-domain interactions between FAT10 and Ub studied by molecular dynamics simulations.

**Figure supplement 2.** The difference in inter-domain long range contacts between FAT10-Ub and diubiquitin studied by molecular dynamics simulations.

**Figure supplement 3.** 2D-HSQC spectrum with assigned peaks of ubiquitin residues in each of the chimeric constructs: (**A**) Fat10D2-Ub and (**B**) Ub-Fat10D2.

**Figure supplement 4.** Changes in the ubiquitin backbone dynamics by conjugation to Fat10 domains.

**Figure supplement 5.** Thermodynamic Coupling in FAT10-substrate conjugate system.

---

(*Figure 8—figure supplement 3A and B*). Chemical shift perturbations (CSPs) of amide NMR resonances report the change in the chemical environment in a protein. High NMR CSPs were observed in additional regions in Ub when Fat10D2 is conjugated to the C-terminus (*Figure 8C and D*), as expected from MD simulations. We then measured Tc values to estimate Fat10D2's effect on the strength of hbonds in the substrate (*Figure 8—figure supplement 3C*). Differences in averaged Tc values between Fat10D2-ubiquitin and free ubiquitin show that the hbonds in the strands β1β2, helix α1, and the α2 loop in the substrate ubiquitin are weakened (*Figure 8E*). Additional hbonds are destabilized in β3β4 and β5 when Fat10 is conjugated at the C-terminus, which is commensurate with its lower $\Delta G_{unfolding}$. Experimental NMR Tc measurements correlate excellently with the changes in local intra-substrate contacts observed in the MD simulations (*Figure 8E* and *Figure 8—figure supplement 2E, F*). The substrate's conformational entropy was quantified from NMR relaxation order parameters (*Figure 8—figure supplement 4*). Conjugation at the N-terminal increases entropy at the β1β2 loop, β2, and α1 in the substrate (*Figure 8F*). Entropy increases further in these regions when D2 is conjugated at the C-terminus. In addition, the α2 loop and β3β4 regions become more entropic.

Overall, the Fat10 domain contacts the substrate at multiple regions to weaken its local stability and increase disorder. The intrinsic flexibility of the conjugation site regulates Fat10's impact on substrate stability.

The substrate may also influence the stability of the proteasome-targeting tag. The Fat10D2 tag is more compact (low Rog) in free form than in the tag-substrate conjugate (*Figure 8—figure supplement 5A*). Moreover, the Fat10D2 tag unfolds faster when conjugated with the substrate (*Figure 8—figure supplement 5B*), suggesting a thermodynamic coupling between the tag and the substrate. The tag-substrate contacts are greater when the tag is conjugated to ubiquitin C-termini (*Figure 8—figure supplement 1A*). Consequently, the tag unfolded faster when conjugated to the substrate C-termini (*Figure 8—figure supplement 5C*), suggesting that the tag stability also depends on the conjugation site.

## Discussion

Posttranslational modification with proteasome-targeting tags is necessary but insufficient for protein degradation. The substrate/proteasome engagement and the substrate unfolding are rate-limiting steps for degradation (*Bard et al., 2019*; *Matyskiela et al., 2013*). Global disorder, topology, local regions with a high disorder, and biased sequences in the substrate are critical factors that regulate proteasomal degradation (*Correa Marrero and Barrio-Hernandez, 2021*). Whether the structural flexibility of the proteasome-targeting tags impacts substrate degradation is unclear. We report that the proteasome-targeting tag Fat10 has a malleable structure that samples multiple partially unfolded forms at physiological temperature. The lack of strong long-range electrostatic interactions between the central helix α1, the β1β2 strands, and the α2 loop, creates the flexible fold that provides weak resistance to mechanical unfolding. These properties of Fat10 expedite its unfolding, degradation, and degradation of Fat10 substrates, highlighting the role of structural plasticity of proteasome-targeting tags in regulating protein degradation rate. Our data correlate well with the previous observations that substituting Fat10 Ubl domains with ubiquitin impedes proteasomal degradation (*Aichem et al., 2018*).

Fat10's impact on the substrate structure and dynamics is noteworthy, as it destabilized various substrates in both in-vitro and cellular conditions. We chose two ultra-stable proteins, CFP and Ub, whose melting points are above 90 °C. Fat10 reduced the unfolding energies of both substrates by 3–5 kcal/mol, which was considerably higher than the effect of ubiquitin on these substrates. Our simulations suggested that Fat10 makes several nonspecific intermolecular contacts with the substrate to perturb the intra-substrate contacts. Competition between inter- and intra-molecular interactions has been recently highlighted, where critical salt-bridge interactions at protein-protein interfaces are "stolen" by new posttranslational modifications within the interacting proteins (*Skinner et al., 2017*; *Mohanty et al., 2019*). A similar mechanism may disrupt critical intra-substrate contacts by substrate-tag collision. Intriguingly, the substrate reciprocally increases partially unfolded regions in Fat10. Thermodynamic coupling between the substrate and tag functions in tandem to increase partially unfolded regions in the substrate-tag conjugate and accelerate its degradation.

Increased enthalpy and conformational entropy suggest local order-to-disorder transitions in proteins, giving rise to partially unfolded forms. We have measured the Tc values to estimate hbond strength, where higher ΔTc values indicate reduced hbond strength and increased enthalpy. Entropy was calculated from NMR relaxation order parameters. The increase in local enthalpy and entropy values identified Fat10-induced local disorder in the substrate, which agreed well with the computational data. An intriguing question is whether the chemical environment at the Fat10 conjugation site can regulate substrate stability. Our results show that conjugation at disordered regions allows greater conformational space for Fat10, increasing its collisions with the substrate and reducing substrate stability. We explored further by studying the effect of Fat10 conjugation on several substrates with varied electrostatic surface potentials (*Ravichandran and Das, 2024*) and found that Fat10 conjugation on positively charged surfaces has a lower/insignificant effect on substrate energies because the negatively charged Fat10 tag forms stable interactions with the substrate, which reduces its conformational flexibility. These studies suggest that the local environment around the conjugation site can regulate Fat10's effect on the substrate energetics.

Posttranslational modification with ubiquitin destabilizes substrates (*Carroll et al., 2020*; *Hagai and Levy, 2010*; *Carroll et al., 2021*; *Morimoto et al., 2016*). However, the proteasomal degradation

pathways of ubiquitin and Fat10 substrates are distinct. Post cleavage by proteasome deubiquiti-nases, ubiquitin's impact on the substrate is unsustained. Hence, few substrates may not interact with the proteasome ATPases and escape degradation (*Correa Marrero and Barrio-Hernandez, 2021*). Fat10 remains conjugated to the substrates, and its effect on the substrate persists until the conjugate translocates to the ATPases. Since Fat10 is degraded along with the substrate, its malleability directly influences the substrate degradation rate. While the N-termini of ubiquitin is rigid, a short N-terminal disordered region in Fat10 contributes to its degradation (*Aichem et al., 2018*). Moreover, Fat10 and ubiquitin's impact on the substrate structure and energetics are distinct. Fat10 domains have a greater exposed hydrophobic surface than ubiquitin, which can induce more nonspecific substrate-tag colli-sions. Fat10's ground state is in equilibrium with partially unfolded forms with exposed hydrophobic patches, further enhancing interactions with the substrate. Together, these structural and thermody-namic properties of Fat10 create a higher disorder in the substrate than in ubiquitin. These mecha-nistic differences explain the unfoldase(Cdc48)-independent rapid degradation of Fat10 substrates. In hypoxic conditions, however, the 20 S proteasome degrades ubiquitin conjugated to disordered substrates (*Sahu et al., 2021*). Since ubiquitin is uncleaved from the substrate, the mechanistic differ-ences between the Fat10 and ubiquitin for 20 S proteasomal degradation is an intriguing question for further investigation.

The study has a few caveats. Fat10 was conjugated to the N- or C-terminus of the substrates here and not to specific lysines. Without an identified FAT10 E3-substrate pair that is active under in vitro conditions, it is technically challenging to conjugate substrate lysines site-specifically. The discovery of novel FAT10 E3s will facilitate such studies in the future. Previous computational research suggested that conjugation between two proteins can destabilize their folded state, stabilize their unfolded state, or both (*Bigman and Levy, 2020*; *Sokolovski et al., 2015*). While our NMR experiments have exclusively studied Fat10's effect on the substrate's folded state, further studies are required to reveal its impact on the substrate's unfolded state.

The proteasome degrades 80–90% of cellular proteins (*Jang, 2018*), and understanding its activity has broad implications for understanding cellular response mechanisms to stress and infection. The immune cells undergo major reprogramming of signaling and antigen presentation during inflamma-tion, and the proteasome plays an essential role in this process (*Çetin et al., 2021*). An important question is how proteasome degradation is modulated in these cells and why Fat10 is upregulated during infection. Our work provides a plausible understanding that the upregulation of a malleable proteasome-targeting signal Fat10 helps immune cells modulate the proteasomal degradation rate in response to infection. Fat10 is designed as an efficient proteasomal degradation signal activated as an inflammatory response. Intriguingly, Fat10 also presents scope for designing therapeutics by controlled protein degradation. Understanding the physical and thermodynamic effects of Fat10 modification is essential to engineer it for therapeutics.

## Materials and methods
### Cloning

A *Fat10* gene (Life Technologies) was constructed where C7, C9, C160, and C162 were mutated to alanine. Such cysteine mutants of Fat10 have been used previously for in vitro studies (*Aichem et al., 2018*; *Truongvan et al., 2022*). The thermal stability of cysteine mutant Fat10 is similar to wild-type Fat10 (*Aichem et al., 2018*). The gene was then sub-cloned in a pET3a vector tagged with 6xHis using NdeI and BamHI restriction sites and was used in the in vitro experiments. The N-ter-minal domain (Fat10D1: 1-83aa with the GG motif at the C-terminal end) and the C-terminal domain (Fat10D2: 84-165aa) were PCR amplified and further cloned into pET3a and pET14b vector, respec-tively. Fat10D1 was tagged with 6xHis, whereas Fat10D2 was tagged with 6xHis-scSMT. A precision protease site was introduced between the scSMT tag and Fat10D2.

The CFP plasmid was a generous gift from Dr. Aakash Gulyani, which was further sub-cloned into a pET3a vector using NdeI and BamHI restriction sites with a 6xHis tag at its N-terminus. CFP chimeric constructs Ub-CFP, Fat10-CFP, Fat10D1-CFP, and Fat10D2-CFP were cloned using overlapping PCR (*Bryksin and Matsumura, 2010*), where the amplified product of CFP was inserted at the C-ter-minus of the genes. For the ubiquitin chimeric constructs, Fat10D2 and Fat10D1 were swapped by ubiquitin to make Fat10D1-Ub and Ub-Fat10D2 constructs, respectively. The Fat10D2-Ub was first

created by sub-cloning Fat10D2 in the pGEX6P1 vector with the GST tag and then fusing the PCR product of ubiquitin at the C-termini of Fat10D2. Similarly, ubiquitin was inserted at the C-terminus of Fat10 to create the Fat10-Ub construct. For the tryptophan-based melt studies, phenylalanine (F) at the 45th position of ubiquitin was mutated to tryptophan (W) in Ub, Fat10-Ub, Fat10D2-Ub, and Ub-Fat10D2. The tryptophan residue at the 17th position in Fat10 was mutated to phenylalanine in Fat10-Ub$_{F45W}$ and Fat10D1-Ub$_{F45W}$ by site-directed mutagenesis. All the clones were verified using the Sanger sequencing method.

3xFLAG-wtFat10 (wild type 1–165) was constructed from Life Technologies for the cellular experiments. The gene was further sub-cloned into pcDNA3.1/hyg(+) vector (a kind gift from Dr. Apurva Sarin, NCBS) using HindIII and BamH1 sites. 3xFLAG-wtFat10D1 (aa:1–81) and 3xFLAG-wtFat10D2 (aa:82–165) were further made from 3xFLAG-wtFat10 in pcDNA3.1. After confirming the 3xFLAG-wtFat10D1 clone, a C-terminal tail sequence CYCIGG (same as the C-terminal tail of wtFat10) was inserted. The GG residues in 3xFLAG-wtFat10-GG, 3xFLAG-wtFat10D1-CYCIGG, and 3xFLAG-wtFat10D2-GG were also mutated to non-conjugal AV residues to create 3xFLAG-wtFat10-AV, 3xFLAG-wtFat10D1-CYCIAV, and 3xFLAG-wtFat10D2-AV in the pcDNA3.1 vector. Mutations in Fat10 were performed by site-directed mutagenesis. The Ubk0GV-GFP construct was obtained from Addgene (#11932) (*Dantuma et al., 2006*) and further sub-cloned into pcDNA3.1 to get 3xFLAG-Ub-k0GV. For the cycloheximide assay, GFP chimeric constructs were made by inserting the PCR product of GFP at the C-terminus end of 1xFLAG-UbGV and 3xFLAG-wtFat10AV by PCR. For the *in-cell* protein stability assay, a pseudo wild type CRABP1 gene (with R131Q stabilizing mutation and tetra-Cysteine motif by replacing G102C, D103C substitution followed by further insertion of 2 Cysteine residues between [105]P and [106]K; *Ignatova and Gierasch, 2004*) was constructed from Life Technologies and sub-cloned in pET3a vector with 6xHis tag. The PCR-amplified product of CRABP1 was inserted at the C-terminus end of ubiquitin, and Fat10 constructs were used to make Ub-CRABP1 and Fat10-CRABP1. All primer sequences will be provided upon request to authors by email.

## Protein purification

All the clones were expressed in *E. coli* BL21 (DE3) cells. Post-induction with 1 mM IPTG, the bacterial culture transformed with Fat10, Fat10D1, Fat10D2, Ub-CFP, Fat10-CFP, Fat10D1-CFP, and Fat10D2-CFP were grown at 18 °C for 16 hr, lysed, and resuspended in lysis buffer (Buffer A: 50 mM Tris buffer pH 7.5, 500 mM NaCl, 20 mM Imidazole and 1 mM DTT). The supernatant was loaded onto IMAC (GE) pre-packed column and eluted with imidazole gradient using Buffer A and Buffer B (Buffer A with 500 mM Imidazole). The purified fractions were applied on a size exclusion chromatography on a Superdex 16/600 75 pg column (GE Healthcare) pre-equilibrated with SEC buffer (Buffer C: 50 mM Tris buffer pH 7.5, 250 mM NaCl, with or without 1 mM DTT). All the ubiquitin proteins were also purified, as given above. GST-ppx-Fat10D2-Ub was purified using a GST pre-packed column and eluted with 10–20 mM reduced Glutathione. The purified proteins His-scSMT-Fat10D2, His-scSMT-ppx-Fat10D2-CFP, and GST-ppx-Fat10D2-Ub, were further treated with GST-tagged Precision protease enzyme followed by reverse IMAC. The digested Fat10D2, Fat10D2-CFP, and Fat10D2-Ub were purified by size exclusion chromatography. Ubiquitin and SUMO1 proteins were purified using protocols published elsewhere (*Chatterjee et al., 2019*; *Negi et al., 2020*). For the NMR experiments, Fat10, Fat10D1, Fat10D2, Fat10-Ub, Fat10D1-Ub, Ub-Fat10D2, and Fat10D2-Ub were grown in M9 media containing isotopic Ammonium chloride ($^{15}NH_4Cl$) and/or isotopic Glucose ($^{13}C$-Glucose). The purification protocol was the same as given above.

## Unfolding studies

Chemical denaturation studies by CD spectroscopy were carried out using 20 µM of protein (Ub, SUMO1, Fat10, Fat10D1, and Fat10D2) incubated overnight with various concentrations of Guanidium Chloride (GdnCl, 0–6 M) made in the native buffer at 25 °C. The change in the far-UV signal at 222 nm was monitored using the Jasco J-1500 spectropolarimeter. CFP and its variants were studied similarly by incubating 15 nM of protein with GdnCl. CFP fluorescence was measured using Horiba Fluromax-4 with an excitation wavelength of 434 nm. The emission signal was collected at 474 nm. Ub$_{F45W}$ and its variants were similarly incubated with GdnCl. The signal at 340 nm was monitored for the change in intrinsic tryptophan fluorescence. Chemical denaturation data were normalized to a two-state unfolding equation using SIGMA plot software. The final curve fitting used a monomeric

two-parameter melt equation (*Privalov, 1979*). The calculated free energy of unfolding, $\Delta G_{unfolding}$, and slope of the transition curve, m, were used to calculate free energy ($\Delta G_0$) at 0 M [GdnCl] using the *Equation 1*.

$$\Delta G_{unfolding,\,[GdnCl]} = \Delta G_0 - m[GdnCl].$$

(1)

Proteins were dialyzed to phosphate buffer (50 mM, pH 6.5, 250 mM NaCl) with or without DTT (1 mM) for thermal equilibrium unfolding studies. The unfolding transition was monitored by observing the change in the CD signal at 222 nm on the JASCO J-1500 spectropolarimeter connected to a Peltier. Data were collected from 20°C to 95°C after every 1 °C per minute rise in temperature with 32 s of data integration. Raw data of the CD signal were normalized to a two-state unfolding equation and plotted against temperature. The curve was fitted with a sigmoidal equation to yield $T_m$. Thermal equilibrium unfolding data were analyzed and processed using SIGMA plot software.

## Native-state proteolysis

The protocol for native-state proteolysis has been described previously (*Carroll et al., 2020*). A total of 100 µl aliquots of Ub-CFP and Fat10-CFP were treated with multiple thermolysin protease concentrations (stock concentration 10 mg ml$^{-1}$). At different times, 10 µl of the reaction mixture was taken, quenched with 6x-SDS loading dye, and run on SDS-PAGE gels. An I-Bright imaging instrument captured the in-gel fluorescence image (Life Technologies). Images were inverted to greyscale and quantified by normalizing them to no protease control sample. The quantified data were fitted to the first-order exponential equation and observed proteolysis kinetics ($k_{obs}$) were calculated at different thermolysin concentrations. The mean of $k_{obs}$ (n=3) at different concentrations was fitted to a linear equation against thermolysin concentration. $\Delta G_{proteolysis}$ was calculated using *Equation 3* by using the slope of the linear fit (*Equation 2*). The $k_{cat}/K_M$ of Thermolysin is 99,000 M$^{-1}$ s$^{-1}$ (*Park and Marqusee, 2004*).

$$\text{The slope of } k_{obs} \text{ vs thermolysin} = K_{op}(k_{cat}/K_M)$$

(2)

$$\Delta G_{proteolysis} = -RT \times \ln(K_{op}(k_{cat}/K_M)/99,000)$$

(3)

## NMR studies

$^{13}$C,$^{15}$N isotopically labeled Fat10D1, and Fat10D2 were prepared in 50 mM Tris, 250 mM NaCl buffer with pH 7.5. Standard triple resonance multidimensional NMR experiments $^{15}$N-HSQC, HNCA, HN(CO)CA, CBCACONH, and HNCACB were recorded at 298 K on an 800MHz Bruker Avance III HD spectrometer equipped with a cryoprobe head. These experiments yielded the backbone assignment of Fat10D1 and Fat10D2. For proteins Fat10D1-Ub and Ub-Fat10D2, only the ubiquitin residue peaks were assigned using standard triple resonance experiments HNCA, HNCOCA, CBCACONH, and HNCACB. NMR data were processed in NMRpipe (*Delaglio et al., 1995*) and analyzed by NMRFAM-SPARKY (*Lee et al., 2015*) software. After peak-picking of the backbone experimental data in SPARKY, the peaks were assigned by the PINE server (*Lee et al., 2009*) and confirmed manually.

## Cell culture experiments

Approximately 0.1 million cells/well of a 12-well plate were seeded with HEK293T cell line and transfected after 70% confluency using a Lipofectamine reagent (Promega). For the cycloheximide assay, cells were transfected with 3xFLAG-wtFat10-AV, 3xFLAG-wtFat10D1-CYCIAV, 3xFLAG-wtFat10D2-AV, 3xFLAG-UbK0GV and 3xFLAG-UbK0AV. The transfected cells were incubated at 37 °C for 18–20 hr, treated with Cycloheximide (CHX, final concentration 50 µg/mL), and lysed at different time points. For 6 hr, these cells were treated with MG132 (final concentration 10 µM). 20 mM Tris (pH 7.5), 150OD600 = mM NaCl, 1 X protease inhibitor cocktail, and 1% NP40 were used to lyse the cell pellet. The BCA kit (Thermo) estimated the total protein amount. Similar transfections were carried out for FLAG-UbGV-GFP, FLAG-Fat10AV-GFP, FLAG-diUbGV-GFP, and FLAG-Fat10D1-GFP. After 18 hr of incubation, the cells were treated with CHX (100 µg/mL) in the presence and absence of p97 inhibitor (DBeQ; final concentration 10 µM) and further incubated for different time points. All the Immunoblots were probed with Mouse anti-FLAG antibody (SIGMA, 1 in 10,000 dilutions) and Mouse anti-β-actin antibody (Santa Cruz, 1 in 5000 dilutions) and further probed with HRP conjugated anti-mouse secondary antibody at 1:10,000 dilutions. The blots were developed using clarity ECL (Bio-Rad) staining reagent,

and the chemiluminescence signal was observed in Image Quant LAS 4000 (GE). The Immunoblots for GFP variants were probed similarly with the mouse anti-FLAG antibody. Tubulin was used as the loading control for normalization and probed with a Mouse anti-tubulin antibody (SIGMA, 1 in 3000 dilutions). All the immunoblots of GFP variants were developed, and the chemiluminescence signal was observed in the Amersham Imager 400 (GE). Quantification was done using ImageJ software with three biological replicates for each experiment.

## In-cell protein stability assay

The stability of CRABP1 with/without ubiquitin and Fat10 in the cellular environment was monitored by pre-labeling the *E. coli* BL21 (DE3) cells with FIAsH-EDT$_2$ fluorescent dye (final concentration 150 μM) at $OD_{600}$ = 0.5, followed by incubation till $OD_{600}$ reaches 1. IPTG was added to cultures to induce protein synthesis. After two hours, the culture solution was aliquoted (150 μl) and treated with various concentrations of Urea ranging from 0M to 3M, keeping the final volume at 400 μl. The culture solutions were incubated for 30 min before pelleting and washing with native buffer (10 mM Tris buffer, pH 7.5), followed by the fluorescent measurements (excitation 500 nm and emission 531 nm). The rest of the protocol is the same as described previously (*Ignatova and Gierasch, 2004*).

## High-temperature unfolding simulations

Systems were prepared in a cubic box of TIP3p water, with a minimum distance of at least 50 Å between solute atoms and the box edge. Counter ions were added to neutralize the system. The system setup and equilibration procedures were similar to the unrestrained simulations, with 450 K as the equilibration temperature. Protein unfolding simulations were performed at 450 K for 300 ns. The simulations were performed with ten replicas to obtain an average folding pathway.

## ASMD simulations

Constant force explicit solvent ASMD simulations of Fat10D1, Fat10D2, Ub, full-length Fat10, and Ub$_2$ were carried out. The reaction coordinate is the end-to-end distance between the Cα atoms of the first amino acid and the last amino acid of the respective protein. For simulations consisting of Fat10D1, Fat10D2, and Ub, the proteins were pulled at 1 Å/ns velocity, and for full-length Fat10 and Ub$_2$, the pulling velocity was 5 Å/ns. The reaction coordinate was partitioned into ten equal segments, each with ten trajectories. The system was energy-minimized equilibrated, and the resulting coordinates & velocities were used as starting points for ASMD simulations. The simulations were performed with periodic boundary conditions in the NPT ensemble; electrostatic interactions were computed using the PME (Particle Mesh Ewald) method. Nonbonded interactions were treated with a cutoff of 8 Å.

## Structure-based Phylogenetic analysis

The structures of 10 available UBL PDBs were retrieved from RCSB PDB as tabulated in Fig. S1A. Crystal structures were cleaned by removing co-crystal structures, multiple domains, ions, and water. For NMR structures, the best representative model was used. For Fat10 and ISG15, individual UBL domains were isolated. RMSD matrix and structure-based sequence alignment were obtained for all structures based on multiple superimpositions using the MUSTANG tool (*Konagurthu et al., 2006*). Further, the phylogenetic tree was plotted using aligned sequences in MEGA X (*Kumar et al., 2018*).

## Molecular dynamics simulations

Structures for individual domains of Fat10D1 & Fat10D2 were retrieved from PDB IDs 6GF1 and 6GF2, respectively. The full-length Fat10 structure was modeled using Swiss-Modeller with 6GF1 and 6GF2 as templates (*Waterhouse et al., 2018*). Chain B of 6GF1 was further processed for simulation and called Fat10D1. All water and sulfate molecules were removed from the Fat10D1 crystal structure. 6GF2 was an NMR ensemble structure, and the best representative structure, Fat10D2, was further considered for simulation. For chimeric models, the individual domains of Fat10D1, Fat10D2, and ubiquitin were fused between the N-terminal of one domain and the C-terminal of the other using UCSF Chimera (*Pettersen et al., 2004*). These models were further simulated for 500 ns before using the final structure for unfolding simulations. The salt bridge mutants were designed using the Rotamer function in UCSF Chimera. Systems for Fat10D1, Fat10D2, Fat10, and ubiquitin were prepared with the LEaP program of Ambertools18. Systems were prepared in a cubic box of TIP3p water, with a

minimum distance of at least 12 Å between solute atoms and the box edge. Counter ions were added to neutralize the system.

Parameters describing system topology were based on the Amber ff99SBildn force field (*Lindorff-Larsen et al., 2010*). The systems were first relaxed by energy minimization in two stages using the Sander module of Amber18. In the first stage, water molecules were minimized with restraint on protein, and then the entire system was minimized. The respective systems were then heated incrementally in NVT from 0K to 300K for 5 ns with positional restraints (20 kcal/mol/Å²) on protein atoms. Further, system density was equilibrated for 5 ns in the NPT ensemble with positional restraints (20 kcal/mol/Å²) on protein atoms. Further, four subsequent equilibration stages reduced the restraints on the backbone atom from 20 to 0 through a series of molecular dynamics simulations in an NPT ensemble for 400 ps each. The final production run was performed for 2.5 μs in NPT with three replicas. The distance cutoff for short-range nonbonded interactions was set to 1 nm. The particle mesh Ewald (PME) method was used to treat long-range electrostatic interactions. The SHAKE algorithm (*Ryckaert et al., 1977*) was applied to constrain all bonds involving hydrogen atoms. Temperature was set to 300 K using a Langevin thermostat, and pressure was maintained at 1 bar using the Berendsen barostat. Using the hydrogen mass repartitioning (HMR) scheme (*Hopkins et al., 2015*), the integration time step was set to 4 fs. Dynamics were propagated using the leapfrog integrator. Snapshots were saved every 40 ps, giving 65200 conformations from a single run. A total of 7.5 μs (3*2.5 μs) data was pooled for further analysis.

The trajectory analyses were performed using the AMBER suite's CPPTRAJ module (*Hipp et al., 2005*). The averages and standard errors were calculated using in-house scripts and were plotted using the R program. Native contacts were calculated using the native-contacts method in CPPTRAJ with a 7 Å distance cutoff. For backbone native contacts across secondary structure pairs, the native contacts were calculated, defining a 3.5 Å distance cutoff on backbone atoms. The data was averaged across ten individual simulations at 450 K to calculate the mean and standard error. The Free Energy plots obtained from bin populations of the 2-dimensional histograms obtained from binning Root Mean Squared Deviation (RMSD) and Radius of Gyration values using the formula:

$$\Delta G_i = -k\beta * T * \ln(N_i/N_o) \tag{4}$$

where $\Delta G_i$ is the free energy value of bin i, $k\beta$ is Boltzmann's constant in kcal/mol*K, T is the temperature in K, $N_i$ is the population of bin i, and $N_0$ is the population of the most populated bin.

RMSD and Rgyr were calculated using rms and rog methods available in CPPTRAJ with their respective crystal structure as a reference. The data were obtained from three individual 2.5 μs repeats at room temperature. Rgyr represents the Radius of Gyration for the protein without loops. The loops were omitted during the Free Energy calculation to avoid false positives. Rog represents the Radius of Gyration for the entire protein. Principle Component Analysis was performed on Ub, Fat10D1, Fat10D2, and their respective salt bridge mutants. PCA was also performed for Fat10D1-Ub, Fat10D2-Ub, and Ub-Fat10D2 using the Bio3D package *Grant et al., 2006* in *R Development Core Team, 2021*. PDB format trajectory was further produced that interpolates between the most dissimilar structures in the distribution along PC1. To obtain contact maps, per-residue native and non-native contacts were calculated from trajectories, normalized, and plotted using GNUplot.

## Far-UV Circular Dichroism (CD) measurements

The Far-UV CD data was collected using Jasco J-1500 spectropolarimeter in a 1 mm cylindrical quartz cuvette, using a 1 nm/s scan speed from 200 to 250 nm with a digital integration time of 4 s. Five scans at 25 °C were averaged and plotted using the SIGMA plot after converting the mdeg to MRE (Mean Residual Ellipticity). Appropriate buffer scans were subtracted from the protein's scan.

## NMR relaxation measurements

For the studies of the Fat10 dynamics, uniformly labeled ¹⁵N-Fat10, ¹⁵N-Fat10D1, ¹⁵N-Fat10D1-Ub, ¹⁵N-Fat10D2-Ub, ¹⁵N-Ub-Fat10D2, and ¹⁵N-ubiquitin were dialyzed in 50 mM Sodium phosphate buffer, 250 mM NaCl with pH 7.5. The NMR relaxation data were collected at 298 K on an 800 MHz and 600 MHz NMR spectrometer (Bruker). Longitudinal (T1), Transverse (T2) time constraints, and hetero-nuclear Overhauser Enhancement (hetNOE) experiments were carried out using standard pulse sequences in Bruker. For T1 measurements, data were recorded at the following relaxation

delays: 0.004, 0.03, 0.06, 0.1, 0.15, 0.2, 0.4, and 0.8 s. For T2 measurements, the relaxation delays were set to 0.004, 0.017, 0.035, 0.051, 0.068, 0.086, 0.102, 0.119, and 0.136 s. A 5 s pre-saturation was used in the $^{15}$N-Heteronuclear NOE experiment. The reference experiment was carried out with 10 s delay without pre-saturation. The T1-relaxation delays for $^{15}$N-Fat10 were: 0.01, 0.025, 0.05, 0.075, 0.1, 0.2, 0.3 s, whereas the T2 delay time was: 0.01, 0.02, 0.4, 0.06, 0.08, 0.1 s and the hetNOE was recorded using standard pulse sequence in Bruker with 5 s as mixing time. The hetNOE was calculated as the ratio of intensities $I_{sat}/I_{ref}$. The hetNOE error was calculated as (NOE err/NOE) = $[(I_{ref}\ err/I_{ref})^2+(I_{sat}\ err/I_{sat})^2]^{1/2}$, where the error of each intensity measurement is the rmsd noise of each plane. Order parameter ($S^2$) was calculated using T1, T2, and hetNOE experiments in RELAX software (*d'Auvergne and Gooley, 2008a*; *d'Auvergne and Gooley, 2008b*) using the Lipari Szabo model-free approach where the $S^2$, correlation time ($\tau_e$), and $R_{ex}$ were obtained. The $S^2$ values were converted to backbone conformational entropy using the entropy meter approach (*Sharp et al., 2015*; *Zeng et al., 2017*).

The temperature dependence of chemical shifts was measured for uniformly labeled $^{15}$N-Ub, $^{15}$N-Fat10D1, and $^{15}$N-Fat10D2 in 50 mM PO4 buffer with 250 mM NaCl at pH 7.5 on the 600MHz spectrometer. The temperature varied from 283K to 313K. The same was measured for $^{15}$N-Fat10, $^{15}$N-Fat10-Ub, and $^{15}$N-ubiquitin at the 800MHz spectrometer. Similar experiments were carried out in 50 mM PO4 buffer with 250 mM NaCl at pH 6.5 for the proteins $^{15}$N-Fat10D1-Ub, $^{15}$N-Fat10D2-Ub, and $^{15}$N-Ub-Fat10D2. The temperature varied from 283K to 323K. The chemical shift of water was used for reference and calibrated using a temperature-independent 4,4-dimethyl-4-silapentane-1-sulf onic acid (DSS) signal. All the NMR HSQC experiments were processed by NMRpipe and analyzed in SPARKY. The chemical shifts in the $^1$H-dimension were analyzed using linear regression by MATLAB. The linear fit between the $\Delta\delta^{NH}$ and temperature provided the temperature coefficient of a residue, Tc = ($\Delta\delta^{NH}/\Delta T$). The Residual Sum Square (RSS) was calculated as $RSS = \sum_{i}^{N} (y_i - f(xi))^2$, where $y_i$ is the i$^{th}$ measured value and $f(x_i)$ is the fitted value of $y_i$.

## Acknowledgements

This work was supported by the Tata Institute of Fundamental Research, Department of Atomic Energy, Government of India, under project identification no RTI 4006, and Science and Engineering Research Board, under project identification no CRG/2021/006032. The NMR data were acquired at the NCBS-TIFR NMR Facility, supported by the Department of Atomic Energy, Government of India, under project no RTI 4006. The NMR facility is also partially supported by the Department of Biotechnology, India, under project number dbt/pr12422/med/31/287/2014. The authors would like to thank Andy Byrd (NCI), Jess Li (NCI), and Kylie Walters (NCI) for reagents and helpful discussions. HN acknowledges a fellowship from UGC-CSIR, India. AR and PD acknowledge scholarships from the Department of Biotechnology, India.

## Additional information

### Funding

| Funder | Grant reference number | Author |
| --- | --- | --- |
| Tata Institute of Fundamental Research | RTI4006 | Ranabir Das |
| Science and Engineering Research Board | CRG/2021/006032 | Ranabir Das |
| Department of Biotechnology, Ministry of Science and Technology, India | dbt/pr12422/med/31/287/2014 | Ranabir Das Aravind Ravichandran |
| Council of Scientific and Industrial Research, India | | Hitendra Negi |

| Funder | Grant reference number | Author |
|---|---|---|

The funders had no role in study design, data collection and interpretation, or the decision to submit the work for publication.

## Author contributions
Hitendra Negi, Conceptualization, Formal analysis, Investigation, Writing - original draft, Writing - review and editing; Aravind Ravichandran, Formal analysis, Investigation, Writing - original draft, Writing - review and editing; Pritha Dasgupta, Shridivya Reddy, Formal analysis, Investigation; Ranabir Das, Conceptualization, Supervision, Funding acquisition, Writing - original draft, Project administration, Writing - review and editing

## Author ORCIDs
Hitendra Negi http://orcid.org/0000-0001-8476-3926
Aravind Ravichandran http://orcid.org/0000-0003-1701-5535
Ranabir Das https://orcid.org/0000-0001-5114-6817

## Decision letter and Author response
Decision letter https://doi.org/10.7554/eLife.91122.sa1
Author response https://doi.org/10.7554/eLife.91122.sa2

## Additional files

### Supplementary files
• MDAR checklist

### Data availability
The manuscript and supporting file include all data generated or analyzed during this study.

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

# Appendix 1

**Appendix 1—key resources table**

| Reagent type (species) or resource | Designation | Source or reference | Identifiers | Additional information |
|---|---|---|---|---|
| Strain, strain background (*E. coli*) | DH5α | Invitrogen | Cat# 18265017 | Plasmid DNA ampilification |
| Strain, strain background (*E. coli*) | BL21(DE3) | Invitrogen | Cat# EC0114 | Protein Expression |
| Cell line (Homo-sapiens) | HEK293T | NCBS, India | | Gift from Dr. Apurva Sarin Lab, NCBS |
| Antibody | Anti-Flag (Mouse monoclonal) | SIGMA | Cat# F3165 | WB (1:10,000) |
| Antibody | Anti-Tubulin (Mouse monoclonal) | SIGMA | Cat# T6199 | WB (1: 3000) |
| Antibody | Anti-β Actin (Mouse monoclonal) | Santa Cruz | Cat# sc47778 | WB (1: 5000) |
| Antibody | HRP-conjugated anti-mouse (goat polyclonal) | SIGMA | Cat# 12–349 | WB (1:10,000) |
| Recombinant DNA reagent | Fat10 (pET3a plasmid) | Life Technologies | | C7A, C9A, C160A, and C162A (See Materials and Methods, Cloning section) |
| Recombinant DNA reagent | Fat10D1 (1-83aa; pET3a plasmid) | This paper | | C7A, C9A, (See Materials and Methods, Cloning section) |
| Recombinant DNA reagent | Fat10D2 (84-165aa; pET14b plasmid) | This paper | | C160A, and C162A (See Materials and Methods, Cloning section) |
| Recombinant DNA reagent | CFP (pET3a plasmid) | This paper | | (See Materials and Methods, Cloning section) |
| Recombinant DNA reagent | Ub-CFP (pET3a plasmid) | This paper | | (See Materials and Methods, Cloning section) |
| Recombinant DNA reagent | Fat10-CFP (pET3a plasmid) | This paper | | (See Materials and Methods, Cloning section) |
| Recombinant DNA reagent | Fat10D1-CFP (pET3a plasmid) | This paper | | (See Materials and Methods, Cloning section) |
| Recombinant DNA reagent | Fat10D2-CFP (pET14b plasmid) | This paper | | (See Materials and Methods, Cloning section) |
| Recombinant DNA reagent | Fat10-Ub (pET3a plasmid) | This paper | | (See Materials and Methods, Cloning section) |
| Recombinant DNA reagent | Fat10D1-Ub (pET3a plasmid) | This paper | | (See Materials and Methods, Cloning section) |
| Recombinant DNA reagent | Ub-Fat10D2 (pET3a plasmid) | This paper | | (See Materials and Methods, Cloning section) |
| Recombinant DNA reagent | Fat10D2-Ub (pGEX6P1 plasmid) | This paper | | (See Materials and Methods, Cloning section) |
| Recombinant DNA reagent | Ub (pET3a plasmid) | This paper | | F45W (See Materials and Methods, Cloning section) |

*Appendix 1 Continued on next page*

*Appendix 1 Continued*

| Reagent type (species) or resource | Designation | Source or reference | Identifiers | Additional information |
|---|---|---|---|---|
| Recombinant DNA reagent | $^{W17F}$Fat10-Ub$_{F45W}$ (pET3a plasmid) | This paper | | W17F in Fat10; F45W in Ub (See Materials and Methods, Cloning section) |
| Recombinant DNA reagent | $^{W17F}$Fat10D1-Ub$_{F45W}$ (pET3a plasmid) | This paper | | W17F in Fat10D1; F45W in Ub (See Materials and Methods, Cloning section) |
| Recombinant DNA reagent | Fat10D2- Ub$_{F45W}$ (pGEX6P1 plasmid) | This paper | | F45W in Ub (See Materials and Methods, Cloning section) |
| Recombinant DNA reagent | Ub$_{F45W}$ -Fat10D2 (pET3a plasmid) | This paper | | F45W in Ub (See Materials and Methods, Cloning section) |
| Recombinant DNA reagent | Ubm1(pET3a plasmid) | This paper | | E34R (See Materials and Methods, Cloning section) |
| Recombinant DNA reagent | Ubm2 (pET3a plasmid) | This paper | | E34R, K27D (See Materials and Methods, Cloning section) |
| Recombinant DNA reagent | CRABP1 (pET3a plasmid) | Life Technologies | | (See Materials and Methods, Cloning section) |
| Recombinant DNA reagent | Ub-CRABP1 (pET3a plasmid) | This paper | | (See Materials and Methods, Cloning section) |
| Recombinant DNA reagent | Fat10-CRABP1 (pET3a plasmid) | This paper | | (See Materials and Methods, Cloning section) |
| Recombinant DNA reagent | 3XFLAG-wtFat10AV (pcDNA3.1 plasmid) | Life Technologies | | |
| Recombinant DNA reagent | 3XFLAG-Fat10D1 (pcDNA3.1 plasmid) | This paper | | (See Materials and Methods, Cloning section) |
| Recombinant DNA reagent | 3XFLAG-Fat10D2-AV (pcDNA3.1 plasmid) | This paper | | (See Materials and Methods, Cloning section) |
| Recombinant DNA reagent | 3XUb-K0AV (pcDNA3.1 plasmid) | This paper | | (See Materials and Methods, Cloning section) |
| Recombinant DNA reagent | 3XUb-K0GV (pcDNA3.1 plasmid) | Life Technologies | | |
| Recombinant DNA reagent | 1XFLAG UbGV-GFP (pcDNA3.1 plasmid) | This paper | | (See Materials and Methods, Cloning section) |
| Recombinant DNA reagent | 1XFLAG diUbGV-GFP (pcDNA3.1 plasmid) | This paper | | (See Materials and Methods, Cloning section) |
| Recombinant DNA reagent | 1XFLAG FAT10AV-GFP (pcDNA3.1 plasmid) | This paper | | (See Materials and Methods, Cloning section) |
| Recombinant DNA reagent | 1XFLAG FAT10D1-GFP (pcDNA3.1 plasmid) | Life Technologies | | |
| Commercial assay or kit | Plasmid DNA purification kit | Promega | Cat# A4160 | Plasmid DNA isolation |
| Commercial assay or kit | PCR clean up kit | Promega | Cat# A9282 | PCR clean up |
| Chemical compound, drug | CHX; MG132 | SIGMA | | |

*Appendix 1 Continued on next page*

*Appendix 1 Continued*

| Reagent type (species) or resource | Designation | Source or reference | Identifiers | Additional information |
|---|---|---|---|---|
| Chemical compound, drug | N15-Ammonium Chloride; C13- D-Glucose | Cambridge Isotope Laboratory., Inc | Cat No# NLM-467; CLM-1396 | Isotope Enrichment media |
| Chemical compound, drug | Protease inhibitor Cocktail | SIGMA | Cat# 11873580001 | |
| Chemical compound, drug | DBeQ | MedChem express | Cat# HY-15945 | |
| Chemical compound, drug | FlAsH-EDT2 | Cayman Chemicals | Cat# 20704 | |
| Chemical compound, drug | ECL reagent | Biorad | Cat# 1705060 | For developing PVDF membrane |
| Chemical compound, drug | BCA kit | Thermo Fisher Scientific | Cat# 23225 | For total protein estimation |
| Software, algorithm | NMR Pipe; Sparky; GraphPad Prism; SIGMA Plot; ImageJ; AMBER | *Delaglio et al., 1995*; *Lee et al., 2015*; *Schmidtke et al., 2014* | | |

