## [Editor Report]

This manuscript probes the ways in which a protein tag might influence the structure, dynamics and stability of a covalently-attached substrate protein. These important findings are of significance to several fields, particularly in understanding how these influences control the abundance of proteins within a cell. The evidence provided is solid and the manuscript will be of interest to scientists working on protein folding and cellular degradation.

---

## [Decision Letter]

**Decision letter after peer review:**

Thank you for submitting your article "Plasticity of the proteasome-targeting signal Fat10 enhances substrate degradation" for consideration by *eLife*. Your article has been reviewed by 2 peer reviewers, and the evaluation has been overseen by a Reviewing Editor and Volker Dötsch as the Senior Editor.

Essential revisions (for the authors):

1. UbiquitinK0 construct ends with GV whereas the FAT10 constructs end with AV. AX is a known C-end degron, which may affect the degradation of FAT10 constructs. For comparison, the authors should have used the same ending sequences for both ubiquitin and FAT10.

2. The in vitro purification of FAT10 constructs possess mutations on residues C7A, C9A, C160A and C162A whereas for cellular studies the wt FAT10 construct has been used. The author should have tested both mutant and wt in parallel for cellular studies to confirm that both mutant and wt behave the same way in cellulo. It remains unclear what role these cysteines play in the thermal stability of FAT10 constructs.

3. Since Ubiquitin can form chains and it is the polyubiquitin chain that is recognized by the proteasome, it is not plausible to compare single ubiquitin with diUb-like FAT10. For the MD-based unfolding studies the authors have used diUb for comparison but for the other experiments, single ubiquitin was chosen. Since FAT10 cannot form chains, it is ideal to compare FAT10 with diUb and should have been tested for in vitro unfolding and cellular degradation experiments. For cellular experiments, it would have been interesting to see the comparison of monoUb with FAT10D1 or FAT10D2 tags as compared to FL FAT10.

4. When we compare the GdnCl melt curves of Ub, FAT010D1 and FAT10D2 with the ASMD-based unfolding shown in Figure 2 we find differences in the results for unfolding. I wonder if the authors tried to check the melting in the absence of the chemical denaturation to see if they could observe any melting at high temperatures. This could be correlated with the observations of the ASMD-based unfolding study.

5. Line 544-550 / Figure 5: Was unmodified CRABP1 unfolding by urea monitored by a 0-3M urea titration as the Ub- and FAT10-modified proteins were? This would be a particularly useful control to show in Figure 5, and to complement the work shown in Figure 6 for CFP.

6. Line 581-583 / 697-700: The manuscript is somewhat confusing on the specificity of Ub/FAT10-interactions with substrates and their effects on substrate specificity. The CFP work (581-583) is non-specific, but the Ub work (697-700) suggests differences that are quite dependent on attachment point of the tag onto the substrate. While perhaps not truly "specific" per se, the attachment point dependence suggests some degree of specificity.

*eLife*'s editorial process also produces an assessment by peers designed to be posted alongside a preprint for the benefit of readers.*Reviewer #1 (Recommendations for the authors):*

In general, I encourage a review of the presentation – this was a challenging manuscript to review given the extensive use of supplementary information (could some of these data be moved into the main work or removed?). Figure legends varied in format substantially across the manuscript on account of their level of detail or summary; some were quite clear (e.g. Figure 2) while others less so (Figure 6).

Line 505-507: needs to be explicitly stated that these are 15N relaxation parameters.

Line 544-550 / Figure 5: Was unmodified CRABP1 unfolding by urea monitored by a 0-3M urea titration as the Ub- and FAT10-modified proteins were? This would be a particularly useful control to show in Figure 5, and to complement the work shown in Figure 6 for CFP.

Line 581-583 / 697-700: The manuscript is somewhat confusing on the specificity of Ub/FAT10-interactions with substrates and their effects on substrate specificity. The CFP work (581-583) is non-specific, but the Ub work (697-700) suggests differences that are quite dependent on attachment point of the tag onto the substrate. While perhaps not truly "specific" per se, the attachment point dependence suggests some degree of specificity.

*Reviewer #2 (Recommendations for the authors):*

1. UbiquitinK0 construct ends with GV whereas the FAT10 constructs end with AV. AX is a known C-end degron, which may affect the degradation of FAT10 constructs. For comparison, the authors should have used the same ending sequences for both ubiquitin and FAT10.

2. The in vitro purification of FAT10 constructs possess mutations on residues C7A, C9A, C160A and C162A whereas for cellular studies the wt FAT10 construct has been used. The author should have tested both mutant and wt in parallel for cellular studies to confirm that both mutant and wt behave the same way in cellulo. It remains unclear what role these cysteines play in the thermal stability of FAT10 constructs.

3. Since Ubiquitin can form chains and it is the polyubiquitin chain that is recognized by the proteasome, it is not plausible to compare single ubiquitin with diUb-like FAT10. For the MD-based unfolding studies the authors have used diUb for comparison but for the other experiments, single ubiquitin was chosen. Since FAT10 cannot form chains, it is ideal to compare FAT10 with diUb and should have been tested for in vitro unfolding and cellular degradation experiments. For cellular experiments, it would have been interesting to see the comparison of monoUb with FAT10D1 or FAT10D2 tags as compared to FL FAT10.

4. When we compare the GdnCl melt curves of Ub, FAT010D1 and FAT10D2 with the ASMD-based unfolding shown in Figure 2 we find differences in the results for unfolding. I wonder if the authors tried to check the melting in the absence of the chemical denaturation to see if they could observe any melting at high temperatures. This could be correlated with the observations of the ASMD-based unfolding study.

---

## [Author Response]

Essential revisions (for the authors):1. UbiquitinK0 construct ends with GV whereas the FAT10 constructs end with AV. AX is a known C-end degron, which may affect the degradation of FAT10 constructs. For comparison, the authors should have used the same ending sequences for both ubiquitin and FAT10.

Thanks for the relevant suggestion. We have now performed the assay with UbK0AV and compared it with Fat10AV (Figure 1E). The corresponding gel data are plotted in Figure 1 Supplemental 1E. With the same ending sequences, the Ub is stable compared to Fat10, which supports the study's conclusion.

2. The in vitro purification of FAT10 constructs possess mutations on residues C7A, C9A, C160A and C162A whereas for cellular studies the wt FAT10 construct has been used. The author should have tested both mutant and wt in parallel for cellular studies to confirm that both mutant and wt behave the same way in cellulo. It remains unclear what role these cysteines play in the thermal stability of FAT10 constructs.

For the cellular studies, we preferred to use the wild-type Fat10. For in-vitro studies, due to solubility issues, we used the Cys-to-Ala mutant Fat10. The Cys-mutant is commonly used to study Fat10 under in-vitro conditions and has been utilized by other laboratories [Truongvan, N., Li, S., Misra, M. et al. Structures of UBA6 explain its dual specificity for ubiquitin and FAT10. Nat Commun 13, 4789 (2022); Aichem, A., Anders, S., Catone, N. *et al.* The structure of the ubiquitin-like modifier FAT10 reveals an alternative targeting mechanism for proteasomal degradation. *Nat Commun* 9, 3321 (2018)].

Given that our study shows that Fat10 has lower thermal stability than Ub, there may be a concern about whether cysteine mutations are causing the difference in stability. However, previous research has shown that the cysteine mutant has a comparable melting temperature with wild-type Fat10 (wild-type Tm=41°C, Cys-mutant Tm=47°C, ∆Tm=6°C), suggesting that the cysteines have a negligible role in the thermal stability of Fat10 [Aichem, A., Anders, S., Catone, N. *et al.* The structure of the ubiquitin-like modifier FAT10 reveals an alternative targeting mechanism for proteasomal degradation. *Nat Commun* 9, 3321 (2018)]. In comparison, the ∆Tm between ubiquitin and fat10 is 40C, which is significantly higher. Moreover, the wild-type Fat10 has lower stability than the cysteine mutant, which supports our conclusion. Repeating the cellular experiments with the cysteine-mutant may not be essential, as the thermal stability of the cysteine-mutant and wild-type Fat10 are comparable. We have now mentioned this point with appropriate references in the text in page 38, lines 3-4.

3. Since Ubiquitin can form chains and it is the polyubiquitin chain that is recognized by the proteasome, it is not plausible to compare single ubiquitin with diUb-like FAT10. For the MD-based unfolding studies the authors have used diUb for comparison but for the other experiments, single ubiquitin was chosen. Since FAT10 cannot form chains, it is ideal to compare FAT10 with diUb and should have been tested for in vitro unfolding and cellular degradation experiments. For cellular experiments, it would have been interesting to see the comparison of monoUb with FAT10D1 or FAT10D2 tags as compared to FL FAT10.

Thanks for the suggestions. We have now compared the stability of diUb-GFP and Fat10-GFP in cells (Figure 5A). Moreover, we have also compared the stability of monoUb-GFP with Fat10D1-GFP in cells (Figure 5B). These measurements confirm a substantial change in the substrate's stability when conjugated to Fat10.

Unfortunately, cloning a diUb-CFP for in-vitro studies was challenging for some technical reasons. However, we can compare the monoUb-CFP, Fat10D1-CFP, and Fat10D2-CFP melts in Figure 6A to note a substantial effect of Fat10 domains on the substrate’s stability compared to Ub. We have changed the text in the page 24, line 14 to reflect this point.

4. When we compare the GdnCl melt curves of Ub, FAT010D1 and FAT10D2 with the ASMD-based unfolding shown in Figure 2 we find differences in the results for unfolding. I wonder if the authors tried to check the melting in the absence of the chemical denaturation to see if they could observe any melting at high temperatures. This could be correlated with the observations of the ASMD-based unfolding study.

In the GdnCl experiments, Ub had higher ∆Gunfolding and greater stability than Fat10D1 and Fat10D2. When we performed melting experiments without any chemical denaturant, Ub again showed higher stability than Fat10 domains (Figure 1J, last column). If we understand correctly, this is the experiment the reviewer has suggested.

In the ASMD, Ub required more work to unfold than Fat10 domains, suggesting that Ub has higher mechanical stability than Fat10. Overall, the chemical melt, temperature melt, and ASMD results are in sync, suggesting higher stability in Ub.

The difference in unfolding energy between Ub and Fat10 domains is distinct in chemical unfolding and mechanical unfolding. It is important to note that comparing the GdnCl melt curves with ASMD-based unfolding results involves contrasting two fundamentally different techniques. Previous studies have reported that the unfolding pathways and the ΔG values obtained from different techniques, such as mechanical unfolding and chemical denaturation, are not directly comparable [Stirnemann G, et al., How force unfolding differs from chemical denaturation. Proc Natl Acad Sci U S A. 2014;111(9):3413-8]. Similarly, mechanical unfolding and temperature melts are not directly comparable [Tapia-Rojo R, et al., Thermal versus mechanical unfolding in a model protein. J Chem Phys. 2019;151(18):185105]. Hence, we have considered the inherent disparities between chemical denaturation, thermal melts, and mechanical unfolding, which is crucial when interpreting the results.

5. Line 544-550 / Figure 5: Was unmodified CRABP1 unfolding by urea monitored by a 0-3M urea titration as the Ub- and FAT10-modified proteins were? This would be a particularly useful control to show in Figure 5, and to complement the work shown in Figure 6 for CFP.

Thanks for the suggestion. We have now provided the in-cell urea-mediated unfolding of CRABP1 as Figure 5 Supplemental 1F. The engineered CRABP1 protein was fairly stable as shown in Figure 5E, but was severely destabilized by Fat10.

6. Line 581-583 / 697-700: The manuscript is somewhat confusing on the specificity of Ub/FAT10-interactions with substrates and their effects on substrate specificity. The CFP work (581-583) is non-specific, but the Ub work (697-700) suggests differences that are quite dependent on attachment point of the tag onto the substrate. While perhaps not truly "specific" per se, the attachment point dependence suggests some degree of specificity.

The reviewer has rightly pointed out that in the substrate ubiquitin when we attached Fat10 to either N- or C-terminus, we observed a site-specific effect of Fat10 conjugation on the substrate’s free energy. While this paper was in review, we investigated this further by studying the site-specific effect of Fat10 conjugation on three distinct substrates. [Ravichandran A, et al., “The Thermodynamic Properties of Fat10ylated Proteins Are Regulated by the Fat10ylation Site,” ACS Omega, 2024,9(20):22265-22276]. Conjugation sites near negatively charged or neutral surfaces in the substrates have a more significant effect on substrate energies than positively charged surfaces. When the site is around a positively charged surface, the negatively charged Fat10 D2 domain forms stable interactions with the surface, reducing the conformational flexibility of the Fat10 tag and its effect on the substrate. This point has now been included in the discussion in page 34, line 15 (in track changes).

Recent studies have studied the specific effects of Ub-conjugation [Carroll EC, et al., Site-specific ubiquitination affects protein energetics and proteasomal degradation. Nat Chem Biol. 2020 Aug;16(8):866-875]. Together, these studies suggest specificity in tag/substrate interactions, which could be a regulatory process to avoid inadvertent degradation.

Reviewer #1 (Recommendations for the authors):In general, I encourage a review of the presentation – this was a challenging manuscript to review given the extensive use of supplementary information (could some of these data be moved into the main work or removed?). Figure legends varied in format substantially across the manuscript on account of their level of detail or summary; some were quite clear (e.g. Figure 2) while others less so (Figure 6).

We thank the reviewer for the comments. Much of the supplementary information for Figures 2, 3, and 7 has been incorporated into the main figures. Figure 7 has been split into two figures for this purpose. A few of the redundant supplementary figures have been removed. The discrepancy between figure legends has been resolved. All figure (including supplementary figure) legends have a summary sentence and are more detailed.

Line 505-507: needs to be explicitly stated that these are 15N relaxation parameters.

This is now explicitly stated.